# Self-loops in evolutionary graph theory: Friends or foes?

**Nikhil Sharma, Sedigheh Yagoobi, Arne Traulsen** *

Department of Theoretical Biology, Max Planck Institute for Evolutionary Biology, Plön, Germany

* traulsen@evolbio.mpg.de

**Data Availability Statement:** All relevant data are within the manuscript. Our code and date is available at https://gitlab.gwdg.de/nsharma/self_loops_egt.

**Funding:** This work was funded by the Max Planck Society. The funders had no role in study design,

## Abstract

Evolutionary dynamics in spatially structured populations has been studied for a long time. More recently, the focus has been to construct structures that amplify selection by fixing beneficial mutations with higher probability than the well-mixed population and lower probability of fixation for deleterious mutations. It has been shown that for a structure to substantially amplify selection, self-loops are necessary when mutants appear predominately in nodes that change often. As a result, for low mutation rates, self-looped amplifiers attain higher steady-state average fitness in the mutation-selection balance than well-mixed populations. But what happens when the mutation rate increases such that fixation probabilities alone no longer describe the dynamics? We show that self-loops effects are detrimental outside the low mutation rate regime. In the intermediate and high mutation rate regime, amplifiers of selection attain lower steady-state average fitness than the complete graph and suppressors of selection. We also provide an estimate of the mutation rate beyond which the mutation-selection dynamics on a graph deviates from the weak mutation rate approximation. It involves computing average fixation time scaling with respect to the population sizes for several graphs.

## Author summary

Evolutionary and ecological dynamics is strongly affected by the underlying population structure. Evolutionary graph theory considers networks in which individuals are placed on the nodes and replace each other via the links. Amplifiers and suppressors of selection are particularly intriguing structures that can effectively change the selective advantage of a mutant compared to unstructured populations. For very low mutation rates, strong amplification requires that mutants can replace their parents via self-loops. We show that this beneficial role of self-loops is reversed when the mutation rate is increased: In this case, self looped-graphs have a lower average fitness in mutation-selection balance. More generally, we show that suppressors of fixation—structures that reduce the fixation of mutants regardless of their relative fitness—can increase the fitness in mutation selection balance both for weak mutation and for strong mutation. This calls for a closer investigation of structures other than the amplifiers of selection.

data collection and analysis, decision to publish, or preparation of the manuscript.

**Competing interests:** The authors have declared that no competing interests exist.

# 1 Introduction

Evolutionary graph theory (EGT) studies the role of spatial structure in evolutionary dynamics [1]. In this framework, a spatially structured population is modelled as a graph with nodes representing asexually reproducing individuals, while the links dictate the interactions among these nodes. In general, the links of a graph can be weighted and directed. So far, the main focus of the EGT has been to study quantities like fixation probability and fixation times for different graphs. The fixation probability is the probability that a mutant individual takes over the population of wild-types, and the time it takes to do so, is called the fixation time. The fixation probability is a central object in evolutionary biology [2–6]. For low mutation rates, it determines the rate of evolution [7, 8]. Based on the fixation probability, most graphs can be categorised into two categories: Amplifiers of selection and suppressors of selection [9]. An amplifier of selection is a structure that—compared to the complete graph (the well-mixed population)—has higher probability to fix beneficial mutants, and lower probability to fix deleterious mutants [10]. On the other hand, a suppressor of selection has higher probability to fix deleterious mutants, and lower probability to fix beneficial mutants than the complete graph.

In a complete graph, every node is alike; therefore, the fixation probability for a mutant starting from any of the nodes is equal. However, this is not true in general. For an arbitrary structure, the fixation probability depends crucially on the node where the initial mutant appears [11, 12]. Hence, the mutant initialisation scheme needs to be specified while stating the fixation probability for a graph. Two commonly used mutant initialisation schemes are uniform mutant initialisation and temperature mutant initialisation. Under the uniform mutant initialisation scheme, the initial mutant is equally likely to appear in every node. Under the temperature mutant initialisation scheme, the initial mutant appears in a node with probability proportional to its temperature, where the temperature of a node is the sum of the weights of the links directed towards the focal node [10]. In general, a graph can have very different fixation probability profiles under different mutant initialisation schemes. For example, the star graph is an amplifier of selection under the uniform mutant initialisation scheme, whereas, it is a suppressor of selection (in the limit of infinite population size) under the temperature mutant initialisation scheme [13].

Recently, evolutionary dynamics on graphs has been studied beyond the fixation time scales by allowing mutations to appear continuously [14–16]. The main quantities of interest in those long-term mutation-selection dynamics are the mutation-selection balance [17] and the mixing time, the time it takes for the dynamics to reach the steady-state [18, 19]. For very low mutation rates, amplifiers of selection attain higher average steady-state fitness in the mutation-selection balance than the well-mixed population, and, suppressors of selection attain lower average steady-state fitness in the mutation-selection balance than the well-mixed population [16]. A suppressor of selection attains lower average steady-state fitness in the mutation-selection balance because it is worse in fixing beneficials mutants and better in fixing deleterious mutants than the complete graph. An amplifier of selection attains higher average steady-state fitness in the mutation-selection balance, because it is better in fixing beneficial mutants and in preventing the fixation of deleterious mutants. In Ref. [20], it has been proven that self-loops are necessary to generate substantial amplification. While we know that in the low mutation rate regime, the self-looped star—an amplifier of selection—adapts better than the complete graph, it is not clear what happens to these self-looped amplifiers when the mutation rate is increased beyond the low mutation rate regime. This is what we investigate here. We find that self-loops can have a detrimental effect on average fitness when the mutation rate increases.

## 2 Methods

### 2.1 Moran Birth-death dynamics with mutation

To study evolutionary dynamics on graphs, we use the Moran Birth-death (Bd) updating. The letter B of the shorthand Bd stands for birth, whereas d stands for death. In Bd updating selection operates during the birth event, and it is represented by the capital letter B. Death occurs randomly with uniform probability, and it is represented by the small letter d. The first letter of a shorthand represents a global event where every individual of the population participates. The second letter of the shorthand represents a local event where only individuals neighbouring the selected individual from the first event participate. More details on the types of evolutionary update rules in spatially structured populations can be found in [21].

To study mutation-selection dynamics on graphs, we use a modified version of Moran Bd updating where mutations occur with probability $\mu$ when an offspring is produced, see Fig 1. One Moran Bd with mutation update step can be described as follows:

1. Birth: First, an individual at node $i$ is selected with probability proportional to its fitness, $\frac{f_i}{\sum_j f_j}$ to reproduce.

2. Mutation: The offspring is identical to the parent individual with probability $1 - \mu$ or a mutant with probability $\mu$. If the offspring is a mutant, its fitness $f'$ is sampled from the mutant fitness distribution $\zeta(f', f_i)$ with $f_i$ being the parent's fitness.

3. Death: A random neighbour of node $i$, say node $k$, is chosen for replacement with probability $\frac{w_{ik}}{\sum_j w_{ij}}$. With probability, $\frac{w_{ii}}{\sum_j w_{ij}}$, the same parent individual can be chosen for replacement via a self-loop. The offspring finally replaces the chosen individual.

An individual's fitness $f$ belongs to a continuous interval bounded by $f_{\min}$ and $f_{\max}$ and remains constant throughout the dynamics as long as it survives. The fitness of an individual is also independent of the frequencies of other types of individuals. In this work, we primarily consider uniform mutant fitness distribution, i.e., $\zeta(f', f) = \frac{1}{f_{\max} - f_{\min}}$. In Sec. 3.5, we investigate evolutionary dynamics with Gaussian mutant fitness distribution.

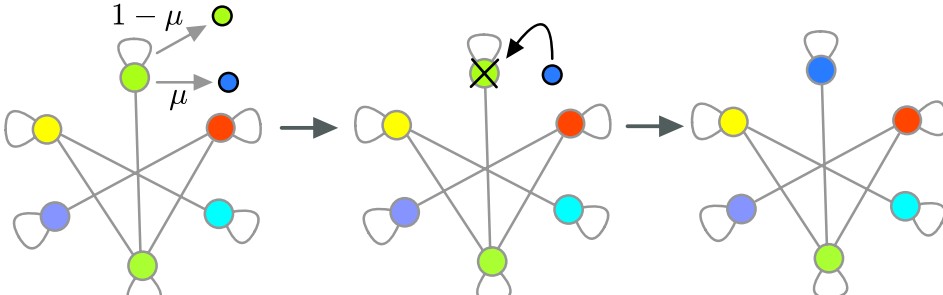

**Fig 1. Birth-death (Bd) updating with mutation.** Here we show an example of the single time step of the Moran Bd updating rule with mutation. First an individual is selected with probability proportional to its fitness to give birth to an offspring. The offspring resembles the parent with probability $1 - \mu$, or mutates with probability $\mu$. If the mutation occurs, the offspring fitness $f'$ is then sampled from the distribution $\zeta(f', f)$ with $f$ being the parent's fitness. In the figure, we have shown the case when mutation takes place. The mutant offspring will then replaces one of the individuals neighboring the parent individual, or the parent individual itself via the self-loop. The choice is made at random with probability proportional to the outgoing weight from the parent node. Here, we have shown the case when the parent individual is replaced by the offspring via the self-loop. The stronger the self-loop, more likely it is for the parent to be replaced by its offspring.

At the level of graphs where each node is occupied by an individual, self-loops were introduced as mathematical objects [13]. However, they make clear sense when each node of a graph is occupied by a population. In that case, a graph is a population of populations [22] where the dynamics in the regime of low migration rate can be interpreted as the dynamics on a graph with strong self-looping [21, 23–27]. However, this work focuses on graphs with one individual per node.

## 3 Results

### 3.1 Amplification in the low mutation rate regime

In this section we briefly summarise the results of Ref. [16] where the Moran Bd mutation-selection dynamics was studied in the low mutation rate regime. In this regime, a newly appeared mutant either reaches fixation or goes extinct before the next mutant appears in the population [28, 29]. For low mutation rates, the population is effectively monomorphic throughout the mutation-selection dynamics and thus, the dynamics can be modelled as a random walk problem on a bounded fitness space where a steady-state is attained in the long run. When a new mutation appears in the population, its fitness $f'$ is sampled from the $\zeta(f', f)$ distribution with $f$ being parent's fitness. Only when the mutation also fixes (which happens with probability $\phi_G^{\mathcal{T}}(f', f)$), the fitness of population $G$ transitions from $f$ to $f'$. Thus, the combined transition rate from point $f$ to $f'$ equal to $\phi_G^{\mathcal{T}}(f', f)\, \zeta(f', f)\mu$. The steady-state for a graph $G$ subjected to a low mutation rate can be computed by assuming detailed-balance [30] as

$$P_G^*(f) = \frac{1}{\int \mathrm{d}f' \, \frac{\phi_G^{\mathcal{T}}(f',f)}{\phi_G^{\mathcal{T}}(f,f')} \cdot \frac{\zeta(f',f)}{\zeta(f,f')}}.$$
(1)

The fixation probabilities entering the steady-state expression 1 are temperature initialised ($\mathcal{T}$), because when a new mutant appears in a homogeneous population, according to the Moran Bd updating stated in Sec. 2.1, it is more likely to appear on the high temperature nodes.

The fixation probability of a mutant with fitness $f'$ on the complete graph with background fitness $f$ is given by [10]

$$\phi_C^{\mathcal{T}}(f',f) = \phi_C(f',f) = \frac{1 - \frac{f}{f'}}{1 - \left(\frac{f}{f'}\right)^N}.$$
(2)

Using the above expression for the fixation probability and the Eq 1, we obtain the average steady-state fitness for the complete graph with uniform mutant fitness distribution,

$$\langle f \rangle_C^* = \int \mathrm{d}f \, f P_C^*(f) = \frac{N}{N+1} \frac{f_{\max}^{N+1} - f_{\min}^{N+1}}{f_{\max}^N - f_{\min}^N}.$$
(3)

Amplifiers of selection attain a higher steady-state average fitness than the well-mixed population. On the other hand, suppressors of selection attain lower steady-state fitness than the well-mixed population, see Ref. [16] for a formal proof. However, a suppressor of fixation, a structure that has lower fixation probabilities than the complete graph regardless of the mutant fitness values, can attain higher average fitness in the mutation-selection balance than the complete graph. This happens because of its ability to reject mutants more efficiently than the complete graph, compensating for its poor ability to fix beneficial mutants. These structures can also attain higher fitness than amplifiers of selection in the steady-state. Therefore, amplifiers

of selection are not the only structures that adapt better than well-mixed populations in the long-term evolutionary dynamics.

## 3.2 Beyond the low mutation rate regime

It has been suggested that the thresholds for mutation rates, beyond which amplifiers of selection deviate from the low mutation approximation are lower compared to the threshold at which the complete graph deviates from the same approximation [8, 31]. This happens because amplification of selection in graphs comes at the cost of higher fixation times of mutants [32, 33]. Thus, amplifiers are more likely to violate the low mutation rate criterion where a mutant appearing in the population should either reach fixation or go extinct before the next mutation appears [8]. However with high fixation times, a new mutant can appear while the previous mutation is still under way towards fixation or extinction, and thus leading to effects like clonal interference [34, 35].

Inside the low mutation rate regime, the steady-state average fitness of the population is independent of the mutation rate. However, the average steady-state fitnesses of various structures are expected to decrease as the mutation rate is increased beyond the low mutation rate regime. Outside the weak mutation rate regime, it is not clear how amplifiers of selection, suppressors of selection, suppressors of fixation, and the well-mixed population are ordered in terms of their average steady-state fitness. To analyse this, we simulate the Moran Birth-death update with mutation for the self-looped star graph (weighted), the complete graph, the cycle graph, the star graph, and the directed line with self-loops. These graphs are shown in Fig 2A. Notice that without self-loops, nodes of the directed graphs that have no incoming links are frozen during the mutation-selection dynamics and their states remain the same throughout the dynamics. To avoid this situation we focus instead on a structure where self-loops are added to all the nodes of the directed line to facilitate their participation in the evolutionary dynamics. The weight matrix of the self-looped star graph is given in Eq 23 with $\lambda$ being the weight of the links directed from leaves to the center and $\delta/(N-1)$ is the weight of the link directed from the center to leaves.

The self-looped star graph is a piecewise amplifier of selection (higher fixation probability for mutations up to a finite fitness advantage) [13] for finite population size. Only in the limit $N \to \infty$, it is a true amplifier of selection. The complete graph, and the cycle graph are isothermal graphs [1]. Under temperature initialisation, for finite $N$, the star graph is suppressor of fixation [13, 16]. The directed line with self-loops is a suppressor of selection [10]. From Fig 2B, we find that in the low mutation rate regime, the steady-state average fitness is highest for the self-looped star graph and the star graph, slightly lower for the complete and the cycle graph, and much lower for the self-looped directed line.

In Ref. [31], it has been shown that the temperature initialised star graph has a lower effective rate of evolution compared to the complete graph. However from Fig 2B, we see that the star graph attains higher steady-state average fitness than the complete graph. Therefore, a structure that speeds up evolution does not necessarily lead to higher fitness in the long-term evolutionary dynamics. Similarly, a structure that slows down evolution does not necessarily lead to lower fitness in the long-term evolutionary dynamics. Although at low mutation rates, the self-looped star graph outperforms all other graphs by attaining the highest steady state fitness, outside the low mutation rate regime, it performs poorly. On increasing the mutation rates, the star with self-loops not only attains lower steady-state fitness than the complete graph, but also lower than the directed line with self-loops, a suppressor of selection. The main reason for this poor adaptation of the self-looped star graph outside the low mutation rate regime are self-loops. We explore this in detail in the following section.

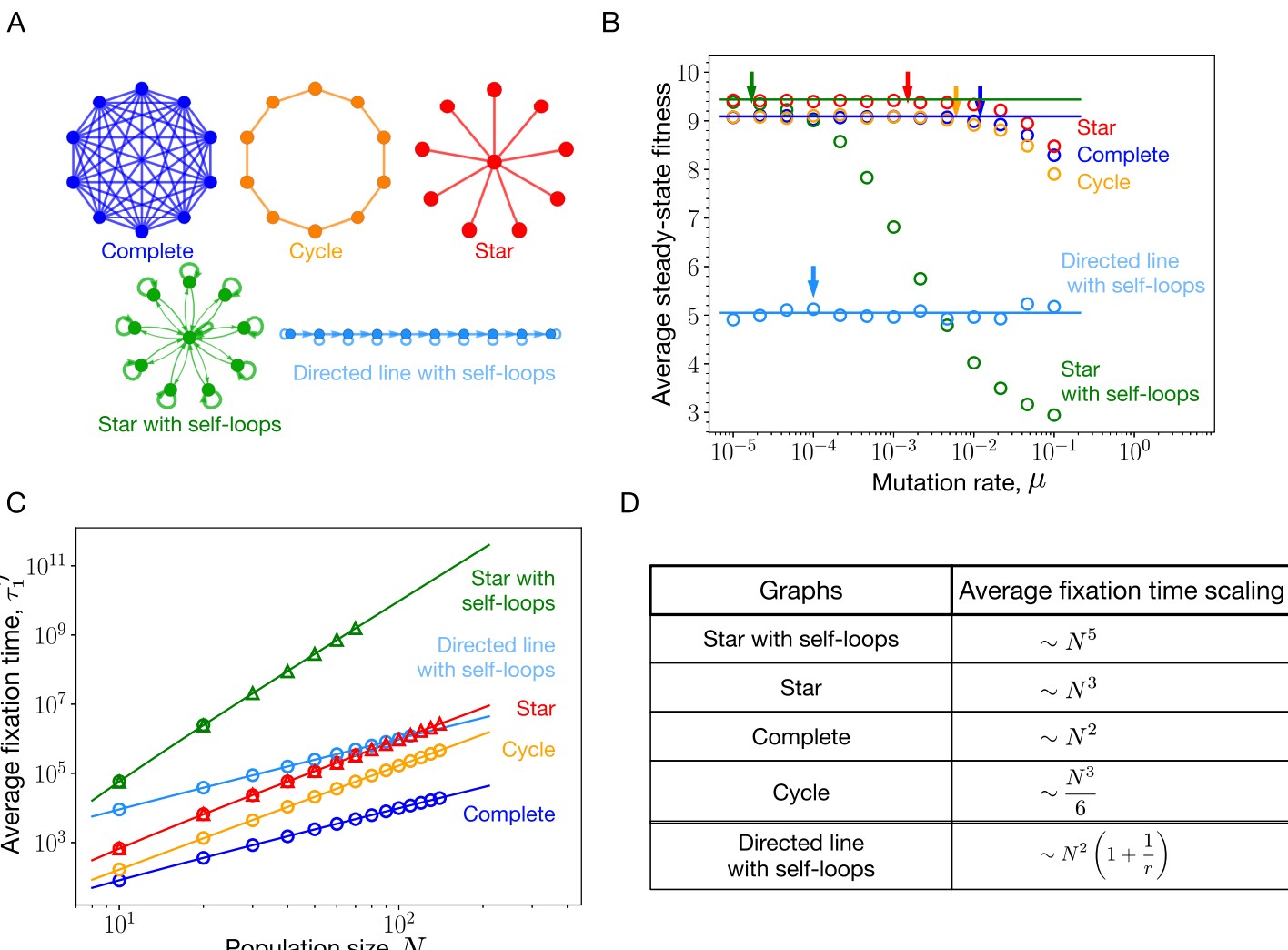

**Fig 2. Mutation rate threshold, $\mu_{th}$.** (A) We mostly work with these five graphs throughout the manuscript. (B) The steady-state average fitnesses obtained using the Moran Birth-death mutation-selection dynamics simulations for the self-looped (weighted) star graph, an amplifier of selection, the star graph, a suppressor of fixation, the self-looped directed line, a suppressor of selection, the cycle graph, an isothermal graph, and the complete graph are shown via circles as a function of mutation rates. Solid horizontal lines represent steady-state average fitnesses for different graphs obtained under the low mutation rate approximation, Eq 1. The arrows mark the mutation rates beyond which the low mutation rate approximation is violated for respective graphs. The graphs with higher average fixation time is expected to deviate earlier, see Eq 4. (C) The average fixation time scaling with $N$ at neutrality is shown for different graphs. Solid lines are the analytical results whereas circles represent Moran Bd simulations. For larger $N$, it gets computationally expensive to work with microscopic Moran Bd simulations, in such cases we use a Gillespie algorithm, shown via triangles. For details on the Gillespie algorithm, refer to App. 5.3.3. (D) The scaling of the average fixation time with population size $N$ for the different graphs. (Parameters: (B) population size, $N = 10$, uniform mutant fitness distribution, i.e., $\zeta(f', f) = \frac{1}{f_{max} - f_{min}}$, (B,C) with 2000 total number of independent realisations used for averaging, $f_{min} = 0.1$ and $f_{max} = 10$).

We conclude this section by providing an estimate for the threshold mutation rate $\mu_{th}$ beyond which the dynamics is considered to be outside the low mutation rate regime. It is given by

$$\frac{1}{\mu_{th}} \approx \max_{r} \left\{ \tau_1^{\mathcal{T}}(r), \tilde{\tau}_1^{\mathcal{T}}(r) \right\}, \tag{4}$$

where $\tau_1^{\mathcal{T}}(r)$ is the average fixation time and $\tilde{\tau}_1^{\mathcal{T}}(r)$ is the average extinction time of a mutant with fitness $r$ relative to the wild-type. Mutants appear according to the temperature

initialisation. Eq 4 follows from the criterion for the dynamics to be in the low mutation rate regime. Recall that the criterion for an evolutionary dynamics to be in the low mutation rate regime is that the time between any two successive mutations should be larger than the time to fixation or extinction (whichever is higher for a given pair of mutant and wild-type fitness) of a mutant. The fixation time and the extinction time of a mutant take random values from specific distributions [36–39]. To arrive at Eq 4, we make an approximation to the criterion by working at the level of average fixation and extinction times. By studying the average fixation and extinction time of the five graphs shown in Fig 2A, except the self-looped directed line, we found the average fixation time of a mutant to be consistently higher than the average extinction time of the mutant, see App. 5.3 for more details. Moreover, it is the average fixation time near neutrality that determines the mutation rate threshold for these graphs. For the complete graph, the phenomenon where the average fixation time peaks near neutrality was discussed in Ref. [40]. For the case of self-looped directed line, we found that the average fixation time decreases as the mutant relative fitness is increased, whereas, the average extinction time increases with increasing mutant fitness, see App. 5.3.5 for more details. However, for a given fitness domain, it is the average fixation time corresponding to the lowest possible mutant's relative fitness that determines the $\mu_{th}$ for the self-looped directed line graph. The star with self-loops has the lowest mutation threshold, since it has the longest average fixation time. At neutrality, for large $N$ the average fixation time for the self-looped star graph scales as $N^5$, whereas for the star graph it scales as $N^3$. The average fixation time scaling for the complete graph is $N^2$, and the average fixation time scaling for the cycle graph is $N^3/6$. For the self-looped directed line graph, the average fixation time scaling which determines $\mu_{th}$ is, $N^2(1 + 1/r)$. The scalings for $\mu_{th}$ for the above-mentioned structures are simply the inverse of the average fixation time scalings mentioned in Tab. D of the Fig 2. The scaling relations are derived in App. 5.3.

## 3.3 Self-loops and high mutation rate regime

Under the Moran Bd update scheme, an offspring always replaces one of the parent's neighbours—unless the parent node is self-looped. For an individual occupying a self-looped node, the offspring can replace the individual with a finite probability. Thus, self-loops effectively decrease the fitness of the parent individual, as the parent cannot spread its offspring freely into the population. The extent of this effect on the parent's fitness depends on the weight of the self-loops. This suggests that the fitness of a highly advantageous strain can be decreased by placing it on a self-looped node with negligible outward flowing weight to the neighbouring nodes [20]. Under Bd updating, the fixation probability of a mutant on a structured population with the weight matrix, $w$, decreases as the diagonal weights of the matrix are increased [41].

For update schemes like bD and dB, and a given structure with the weight matrix $w$, it is necessary to have self-loops ($w_{ii} > 0$) for all $i$, in order for the fixation probability of mutants on that structure to be equivalent to the fixation probability of mutants under a birth death process (of any type) on the self-looped complete graph [41]. Self-loops also fix some issues for the bD and dB dynamics that seem to make them unattractive from a modelling perspective [42]: One problem with the bD updating is that a mutant with fitness tending to zero can have a finite fixation probability. On the other hand, for the dB updating, an infinitely fit mutant can have a fixation probability smaller than one. Self-loops fix these issues.

In order for a structure to be a strong amplifier (a spatial structure where the fixation of a beneficial mutant is guaranteed), self-loops have been proven to be necessary [20], both under the uniform and the temperature initialisation. Though the concept of strong amplifiers is defined for infinite $N$, the self-loops also play a quintessential role in generating amplifiers of

finite $N$ [20]. Intuitively, for a structure to be an amplifier of selection, it should have a sufficient number of cold temperature nodes so that the mutants are less likely to get replaced by wild-type individuals [33] and thus, a mutant type can persist in the population for longer time and spread its offspring into the population. This is where self-loops come into play, they help in creating more of these cold nodes, thereby amplifying selection. Consequently, self-loops contribute substantially in attaining higher fitness in the mutation-selection balance [16].

However as seen in Fig 2B, the steady-state average fitness of the self-looped star, an amplifier of selection, decreases fitness as the mutation rate is increased beyond the mutation threshold. Outside the low mutation rate regime, clonal interference starts to play an important role in the evolutionary dynamics. Therefore, to systematically investigate the effects of self-loops on evolutionary dynamics, we need to analyze the dynamics on structured populations for higher mutation rates. While this can be studied by simulations, it is challenging to obtain analytical insights for arbitrary mutation rates $\mu$. Thus, in addition to simulations we study another—biologically not relevant—extreme of the high mutation rate limit, i.e., $\mu \to 1$. While this seems to be an irrelevant limit, its analysis reveals some crucial properties of evolutionary dynamics that are already relevant for much lower mutation rates.

## 3.4 Sampling fitness from the uniform distribution

In the limit $\mu \to 1$, every time a parent reproduces, the offspring is a mutant. We start with a uniform mutant fitness distribution, $\zeta(f', f) = \frac{1}{f_{\max} - f_{\min}}$ for $f_{\min} \leq f, f' \leq f_{\max}$.

**3.4.1 Reference graph- complete graph with self-loops.** When studying evolutionary dynamics on structured populations, the results are always compared with the dynamics on a reference graph. The standard choice in Evolutionary graph theory for the reference graph is the complete graph (without self-loops). For example, for the case of fixation probabilities and for the mutation-selection dynamics under mutation rates, the complete graph serves as the reference graph. However, for high mutation rates, instead of the complete graph, we choose the self-looped complete graph as a reference. This is because every node of the self-looped complete graph has an equal chance of being replaced by a mutant offspring during every birth event. This also implies that after a sufficiently long time, the states of the nodes would be completely uncorrelated in space and time. The coarse-grained evolutionary dynamics satisfies a master equation where each offspring's fitness $f'$ is chosen randomly from the mutational jump distribution $\zeta(f', f) = \dfrac{1}{f_{\max} - f_{\min}}$ with $f$ being the parent's fitness. The probability density function corresponding to population's state, $P_{SC}(\boldsymbol{f}, t)$ changes as

$$\frac{\mathrm{d}P_{SC}(\boldsymbol{f}, t)}{\mathrm{d}t} = \int \mathrm{d}\boldsymbol{f}' \underbrace{\left(\prod_{i=0}^{N-1} \zeta(f_i, f_i')\right)}_{T_{f \leftarrow f'}} P_{SC}(\boldsymbol{f}', t) - \int \mathrm{d}\boldsymbol{f}' \underbrace{\left(\prod_{i=0}^{N-1} \zeta(f_i', f_i)\right)}_{T_{f' \leftarrow f}} P_{SC}(\boldsymbol{f}, t), \quad (5)$$

where the subscript $SC$ stands for the self-looped complete graph, and, $\boldsymbol{f} = (f_0, f_2 \cdots, f_{N-1})$ is the fitness state of the population of size $N$. By assuming detailed balance [30], i.e.

$$P_{SC}^*(\boldsymbol{f}') = \frac{T_{f' \leftarrow f}}{T_{f \leftarrow f'}} P_{SC}^*(\boldsymbol{f}), \quad (6)$$

and the normalisation condition $\int \mathrm{d}\boldsymbol{f}' P_{SC}^*(\boldsymbol{f}') = 1$, we find the steady-state for the high

mutation rate dynamics on the self-looped complete graph

$$P^*_{SC}(\boldsymbol{f}) = \frac{1}{\prod\limits_{i=0}^{N-1} \int \mathrm{d}f'_i \, \frac{\zeta(f'_i, f_i)}{\zeta(f_i, f'_i)}} = p^*(f_0) \cdot p^*(f_2) \cdots p^*(f_{N-1}).$$

(7)

Here, $p^*(f_i) = \left( \int \mathrm{d}f'_i \, \frac{\zeta(f'_i, f_i)}{\zeta(f_i, f'_i)} \right)^{-1}$ is the marginal probability density function for the node $i$ to have fitness $f_i$. The marginal probability density function also satisfies the normalisation condition $\int \mathrm{d}f \, p^*(f) = 1$. The average steady-state fitness of the self-looped complete graph in terms of the individual node's average steady-state fitness satisfies $\langle f \rangle^*_{SC} = \langle f \rangle^*$, i.e. the average fitness of the population is the same as the average fitness of a node. This follows from the symmetry of the graph. Using the explicit form of the uniform mutational jump density function in Eq 7, we obtain $p^*(f) = \dfrac{1}{f_{\max} - f_{\min}}$, which is independent of $f$. At very high mutation rates, the self-looped complete graph is totally blind to the fitness advantage/disadvantage of a mutant. Therefore, for the self-looped complete graph the average steady-state fitness with $\mu = 1$, and the uniform mutational distribution is $\langle f \rangle^*_{SC} = \dfrac{f_{\max} + f_{\min}}{2}$ which is independent of the population size. Also, the standard deviation of the steady-state fitness is $\dfrac{f_{\max} - f_{\min}}{\sqrt{12N}}$, see Fig 3. For derivation, see App. 5.2. With this, we are now ready to discuss the evolutionary dynamics on various self-looped graphs.

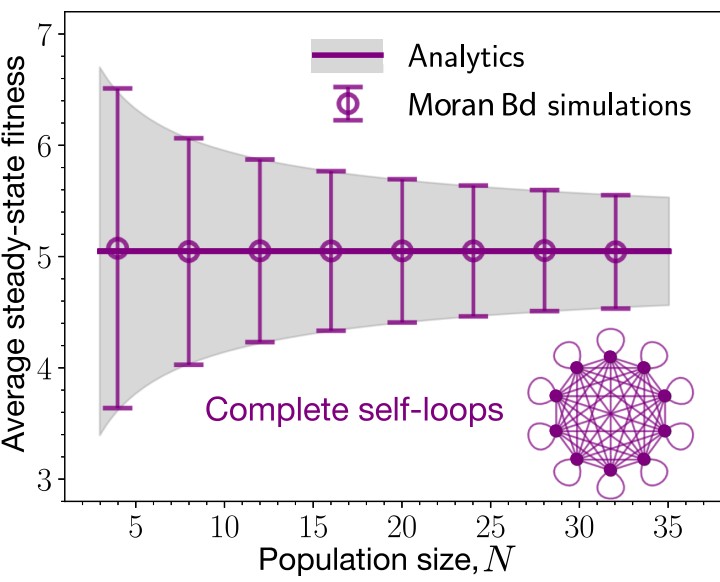

**Fig 3. Reference graph: Complete graph with self-loops.** Here, the mutation-selection dynamics is studied for the self-looped complete graph with $\mu \to 1$. We find a very good agreement for the steady-state statistics between the analytics and the simulations. The thick line represents the analytical average fitness, while the shaded grey area represents the standard deviation around the average. Symbols and error bars show simulations. In the steady-state, on average the self-looped complete graph attains the midpoint of the fitness domain, as the fitness dynamics for each individual node of the population becomes uncorrelated in the fitness space and time. The steady-state average fitness is also independent of the population size. The fluctuations in the steady-state however depends on the population size and decreases with the increase in population size as $1/\sqrt{N}$ (Parameters: $f_{\min} = 0.1$, $f_{\max} = 10$, number of independent realisations is equal to 2000, mutant fitness distribution, $\zeta(f', f) = 1/(f_{\max} - f_{\min})$).

**3.4.2 Self-looped directed line beats the self-looped star.** In this section, we study the high mutation rate dynamics, $\mu \to 1$, on the self-looped directed line and the weighted self-looped star graph. To recall, the self-looped directed line is a suppressor of selection [10, 16], whereas, the (weighted) self-looped star graph is an amplifier of selection. In the low mutation rate dynamics, the self-looped weighted star attains higher steady-state fitness than the self-looped directed line. However, it is unclear what happens in the high mutation rate regime, which is far from a fixation-like dynamics. Simulating the Moran Bd dynamics with $\mu = 1$ for these two graphs, we find that the weighted self-looped star attains lower steady-state fitness not only than the self-looped complete graph, but also in comparison with the self-looped directed line, see Fig 4.

For the case of (weighted) self-looped star graph, from the Fig 4A, all the leaf nodes attain the same steady-state fitness. This is expected due to symmetry reasons. The central node, node 0, stands out, and has the highest fitness. This is because the fitness decreasing effect of the self-loop is minimised by the vast number of outgoing (incoming) links from (to) the central node.

A self-loop affects the node's steady-state fitness depending on the node's connections to other nodes. As an example, the root node 0 of the directed line attains the lowest steady-state fitness among all other nodes, Fig 4B. This is because the only incoming link to node 0 is the self-loop. In a mutation-selection dynamics, a self-loop leads to the decrease in the long-term fitness of a node. This can be understood by the following argument: If a given node is currently occupied by a highly fit individual, it is more likely that during the next Moran Bd update this particular node is selected to reproduce. If this node is self-looped, assuming small outgoing weight to other nodes for now, with high probability the mutated offspring replaces

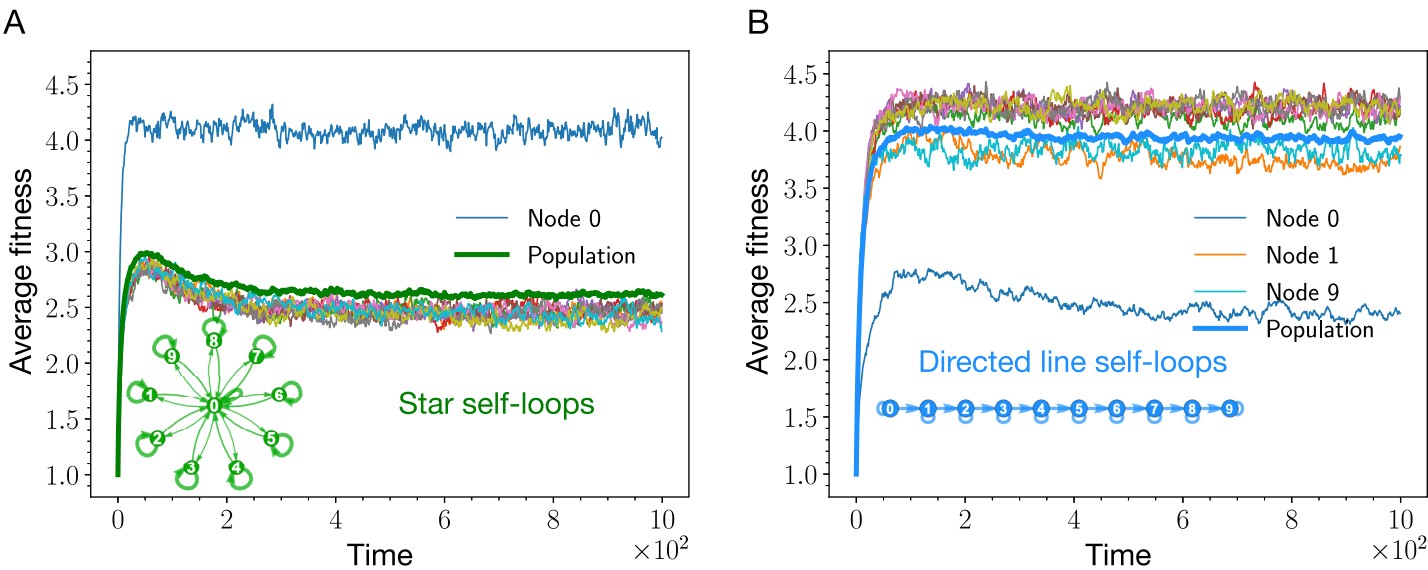

**Fig 4. Nodewise analysis of the star graph with self-loops and the directed line with self-loops.** Here, the average fitness trajectories for each node of the self-looped star graph (shown in panel A) and the self-looped directed line (shown in panel B) are shown. Thick lines represent average fitness trajectories at the population level, whereas, thin lines represent average fitness trajectories for the nodes. The effect of self-loops on a node's fitness depends on the incoming and outgoing weight flowing out of that node. In panel A, self-loops have the least effect on the central node because of relatively higher incoming and outgoing weight. As a result, the central node attains higher average steady-state fitness than the leaf nodes. In panel B, the root node of the directed line has the lowest steady-state average fitness because of the absence of an incoming link to the root node. (Parameters: $N = 10$, $\mu = 1$, $f_{min} = 0.1$, $f_{max} = 10$, number of independent realisations is equal to 2000, mutant fitness distribution, $\zeta(f', f) = \frac{1}{f_{max} - f_{min}}$. For the directed line with self-loops, every outgoing link from a node (including the self-loop) has the same weight. For the self-looped star graphs, the weights of the links follows Eq (23), such that $\lambda = 1/(N-1)$ and $\delta = 1/(N-1)^2$.)

its parent via the self-loop. If the mutated offspring is again very fit, this offspring will again be more likely to be selected to reproduce, and thus, repeating the cycle. This process will repeat until the node's fitness decreases. Therefore, self-loops make it harder for highly fit individuals to persist in the population.

On the other hand, incoming and outgoing links decrease the stated negative effect of self-loops. When a highly fit individual occupying a self-looped node is selected to give birth, its mutated offspring can be placed on a neighbouring node if the parent node has a substantial outgoing weight to other nodes. This decreases the participation of the self-loops in the update process and leads to diminished effects of the self-loops. The role of incoming links is more subtle. Incoming links make a node's fitness state more randomised in accordance with the mutational jump distribution. In the long run, for the case of uniform distribution, the mean of the fitness states attained by an individual node solely via the incoming links is the mid-point of fitness domain. Thus, depending on the mutational fitness jump distribution, incoming links can have beneficial or detrimental effects on a node's fitness. For the case of uniform distribution, compared to self-loops, incoming links have beneficial effect on the population's fitness as adding self-loops decreases the population's fitness below the mid-point of the fitness domain. These arguments explain, why the end node of the directed line has higher steady-state fitness than the root node, but lower fitness than the bulk nodes (node 9 in Fig 4B). The incoming link to the end node decreases the self-loops effect by making the node's fitness more randomised. However, the absence of an outgoing link from the end node makes the negative impact of self-loop still substantial. The steady-state fitness for the node 1 is an interesting case. As a bulk node, its steady-state fitness is lower than other bulk nodes fitnesses. This is because the incoming link to node 1 does not reach its full potential in randomising the fitness. This limitation occurs because the incoming link is activated solely when the root node 0 is selected for reproduction. However since the root node has the lowest fitness, it is less likely to be selected during the update steps.

## 3.5 Sampling fitness from the Gaussian distribution

Until now, in the Moran Bd with mutation update scheme, the mutant's fitness has been sampled from a uniform distribution. However, a lack of correlation between offspring's fitness and parent's fitness is an extreme assumption. Therefore, to examine the robustness of the negative effects of the self-loops observed previously, we study the evolutionary dynamics with the fitness of a mutant offspring sampled from a truncated Gaussian distribution on the fitness domain $[f_{\min}, f_{\max}]$. At a given point of the dynamics, the Gaussian distribution is centered around the parent's fitness with a standard deviation of $\sigma$.

From Fig 5A, we see that adding self-loops decreases the steady-state fitnesses for all the graphs. The effect of adding self-loops is the smallest for the complete graph. This is what we have also observed for the case of uniform mutation fitness distribution. To compare fitnesses, in Fig 5 every self-looped graph has a non self-looped counterpart. For the self-looped directed line, we have the molded directed line which is directed line with an additional link directed from node 1 to the root node (node 0). On increasing the $\sigma$ from 0.1 to 1, compared to other graphs, the self-looped star undergoes a considerable decrease in the steady-state average fitness, Fig 5B. For $\sigma = 1$, the self-looped star graph, an amplifier of selection, attains lower steady-state fitness than the self-looped directed line, a suppressor of selection. For very large $\sigma$, we recover the uniform distribution limit, as expected, see Fig 5C, where all the non-self looped graphs attain the same mutation-selection balance. The average fitness in this case is higher than that of the self-looped complete graph. All self-looped graphs have lower average

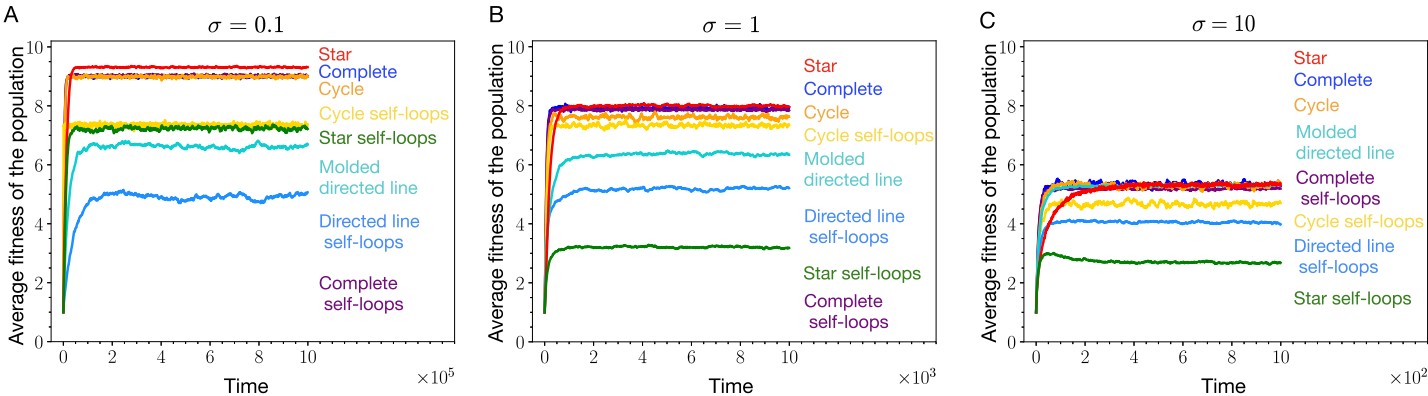

**Fig 5. Sampling mutant's fitness from the Gaussian.** (A) When mutant fitness is sampled from the (truncated) Gaussian distribution with $\sigma = 0.1$, we find that adding self-loops decreases the population fitness in all the graphs. (B) Increasing the $\sigma$ from 0.1 to 1, the average fitness in the steady-state goes down for many graphs. The effect of increasing the $\sigma$ is largest in the heterogeneous star graphs and smallest in the more homogeneous structure like the complete graph. (C) We recover the uniform mutant fitness distribution case for very large $\sigma$, here $\sigma = 10$. In this case, all the non-self looped graphs attain the same steady-state. All self-looped graphs have lower average steady-state fitness than a non-self looped graph and the self-looped complete graph (Parameters: $N = 10$, $\mu = 1$, $f_{\min} = 0.1$, $f_{\max} = 10$, 2000 independent realisations).

fitness compared to the self-looped complete graph. Refer to App. 5.4 for more details on the high mutation rate dynamics for the non self-looped graphs.

Overall, the average steady-state fitness for different graphs increases as $\sigma$ is decreased. This trend agrees with the intuition that low $\sigma$ values provide directionality to the evolutionary dynamics towards higher fitness values. However, not all the self-looped graphs are affected by this directionality equally. The steady-state average fitness for the heterogeneous self-looped graph like, the self-looped star, decreases substantially with increasing $\sigma$, compare Fig 5A and 5B. In contrast, regular self-looped structures like the self-looped complete graph and the self-looped cycle graph, do not experience such a sharp fitness decrease, see again Fig 5A and 5B.

In nutshell, from Fig 5, we conclude that the negative effect of self-loops on the fitness is not limited solely to the uniform mutant fitness distribution.

## 4 Discussion

Amplifiers of selection [1, 10] are fascinating spatial structures. For low mutation rates, these structures can speed up evolution [8] by enhancing the fixation of beneficial mutants. A randomly generated connected spatial structure, under Moran Bd updating with uniform initialisation, is very likely to be an amplifier of selection [9]. Due to their ability to amplify selection and ubiquity, amplifiers of selection have been in the focus of research recently [13, 20, 33, 43–47].

Not only at the fixation time scales but also for the long-term weak mutation rate mutation-selection dynamics, amplifiers of selection perform better than the well-mixed population by attaining higher average fitness [16]. Since mutations occur during reproduction, the fixation probabilities entering the steady-state distribution (mutation-selection balance) for spatial structures are temperature initialised. It has been shown that for structures to amplify selection substantially under temperature initialisation, self-loops are important [20]. However, non self-looped amplifiers of selection with temperature mutant initialisation do exist [48].

In the low mutation rate regime, self-loops help structures perform better but outside the low mutation rate regime, self-looped graphs do not attain higher average fitness than the well-mixed population. In fact, outside the weak mutation rate regime, amplifiers of selection

can even perform worse than suppressors of selection in terms of maximizing fitness. An example is shown in the Fig 2, where the self-looped star graph, an amplifier of selection, attains lower fitness in the mutation-selection balance than the complete and self-looped directed line, a suppressor of selection. To further investigate the effect of self-loops, we have worked in the extreme mutation probability regime, $\mu \to 1$. The idea was to remove other effects from the evolutionary dynamics, and focus solely on the effect of self-loops on the adaptation of a spatially structured population. The insights we obtain working in the high mutation regime can be useful for the intermediate mutation rate regime as well. While we worked with extremely high mutation rates, high mutation rate studies are not uncommon. One of the celebrated theories dealing with high mutation rate mutation-selection dynamics is quasispecies theory [49, 50]. Quasispecies theory is a deterministic theory used to study mutation-selection dynamics in infinite well-mixed population. Its variants have also been used to study finite well-mixed populations [51, 52]. However, the effect of spatial structure remains to be analysed in the quasispecies theory.

While studying mutation-selection dynamics with high mutation rates, the self-looped complete graph naturally serves as the reference graph instead of the complete graph. The fitness dynamics for the self-looped complete graph is random in fitness space and time, i.e., at a given time, the fitness state of a node is independent of its fitness states in the past and the current fitness state of its neighbours. We found that self-loops have a strong fitness-decreasing effect on a node having lower outgoing and incoming weight. In the limit $\mu \to 1$, we found that the non-self-looped graphs attain higher steady-state fitness than their self-looped counterparts. Maybe more surprisingly, all the non-self-looped graphs attain the same average fitness in the mutation-selection balance. All self-looped graphs attain lower steady-state fitness than the complete graph. We also observed the fitness-decreasing effects of the self-loops for the case where the mutant's fitness is sampled from a Gaussian distribution. Thus, the fitness-decreasing effects of the self-loops are not an artefact of a uniform mutant fitness distribution.

We also provide a heuristic measure of the low mutation rate thresholds, $\mu_{th}$, the mutation rate beyond which the evolutionary dynamics is outside the low mutation rate regime. The mutation rate threshold $\mu_{th}$ for a graph depends on the average fixation times and the extinction times of mutants on that graph [8]. As expected, structures with higher fixation times have lower mutation rate thresholds. Therefore, compared to the complete graph and suppressors of selection, amplifiers of selection show deviation from the low mutation rate approximation at lower mutation rates. For a majority of the spatial structures, these thresholds are estimated using the structures' near-neutrality average fixation time scaling with population size. For the directed line with self-loops, the average fixation time grows monotonically with the decrease in mutant's fitness and therefore, $\mu_{th}$ is computed from the average fixation time of a mutant with least possible relative fitness for a fitness domain. In this work, we have derived the large $N$ average fixation time scalings for the star graph, the self-looped star graph and the cycle graph, which in return give the $\mu_{th}$ scalings. Knowing these thresholds one can avoid running heavy simulations deep in the low mutation regime. Since in the low mutation rate regime, the steady-state statistics is independent of the mutation rate, it is sufficient to access the steady-state via simulations by going slightly below the computed mutation rate thresholds but not deep into the low mutation rate regime. Due to higher sojourn times (see Appendix), it is expected for a self-looped graph to have a higher average fixation time for a mutant than its non self-looped counterpart. This however needs a further detailed investigation.

Amplifiers of selection have been in the focus of EGT. However, their promising aspects to optimise fixation of fit mutants are somewhat limited to short-term time scales, where they come with the caveat that they tend to have long fixation times [8, 43]. In the long-term

mutation-selection dynamics, it has been shown in Ref. [16] that suppressors of fixation have the potential to perform better than the amplifiers of selection. This is because of the ability of the suppressor of fixation to reject deleterious mutations more efficiently compensating for its poor probability of fixation for beneficial mutations. Moreover, outside the low mutation rate regime, we see that the temperature initialised star graph, a suppressor of fixation, takes over the self-looped star graph, an amplifier of selection, and maintains higher average fitness in the steady-state throughout the mutation rate regime. However, the reason for the star graph to take over the self-looped star outside the weak mutation rate regime is not clear and requires further investigation. In conclusion, we suggest to broaden the scope of evolutionary graph theory to other structures and to gently move its focus away from amplifiers of selection.

## 5 Appendix

### 5.1 Kolmogorov's Criterion

In the section 3.4.1, we have used the detailed balance condition. Here, we justify the use of detailed balance by proving that the stochastic process at hand is indeed reversible. To do so we make use of Kolmogorov's criterion [53]. According to this criterion, a Markov chain on a fitness space spanned by $f$ is reversible if and only if:

$$T(\boldsymbol{f}_1, \boldsymbol{f}_n) \cdots T(\boldsymbol{f}_3, \boldsymbol{f}_2) T(\boldsymbol{f}_2, \boldsymbol{f}_1) = T(\boldsymbol{f}_1, \boldsymbol{f}_2) T(\boldsymbol{f}_2, \boldsymbol{f}_3) \cdots T(\boldsymbol{f}_n, \boldsymbol{f}_1),\tag{8}$$

for any finite set of ordered fitness states $\boldsymbol{f}_1, \boldsymbol{f}_2, \cdots \boldsymbol{f}_n$.

The basic idea behind the Kolmogorov's criterion relies on the fact that a reversible Markov chain has zero probability current in the steady-state. In our case,

$$T(\boldsymbol{f}, \boldsymbol{f}') \quad = \prod_{i=0}^{N-1} \zeta(f_i, f_i').\tag{9}$$

Since, $\zeta(f_i, f_i') = \frac{1}{f_{\max} - f_{\min}}$, the transition probabilities are independent of fitness. Thus, Kolmogorov's criterion in Eq 8 is satisfied and the Markov chain for the self-looped complete graph presented in the Sec. 3.4.1 is reversible.

### 5.2 Complete graph with self-loops: Fluctuations in the fitness

The standard deviation in the steady-state population fitness for self-looped complete graph is given by $\sqrt{\mathrm{Var}(f)^*}$ where,

$$\mathrm{Var}(f)^* = \langle f^2 \rangle_{SC}^* - (\langle f \rangle_{SC}^*)^2.\tag{10}$$

Here we derive the expression of the second moment for the steady-state of the high mutation rate dynamics of the self-looped complete graph.

$$
\begin{aligned}
\langle f^2 \rangle_{SC}^* \quad &= \frac{1}{N^2} \left\langle \sum_{i=0}^{N-1} \sum_{j=0}^{N-1} f_i f_j \right\rangle^*, \\
&= \frac{1}{N^2} \left( \sum_{i=0}^{N-1} \sum_{j \neq i}^{N-1} \langle f \rangle^* \langle f \rangle^* + \sum_{i=0}^{N-1} \langle f^2 \rangle^* \right), \\
&= \frac{1}{N^2} \left( N(N-1) \langle f \rangle^{*2} + N \langle f^2 \rangle^* \right), \\
&= \left( 1 - \frac{1}{N} \right) \langle f \rangle^{*2} + \frac{1}{N} \langle f^2 \rangle^*,
\end{aligned}
\tag{11}
$$

where $\langle f \rangle^*$ is the average fitness, and $\langle f^2 \rangle^*$ is the second moment of fitness for a node in the steady-state. In the second equality above, we have used that fitness states on a coarse-grained time scale on different nodes of a self-looped complete graph are independently and identically distributed. Therefore the variance in fitness for the population reduces to,

$$
\begin{aligned}
\mathrm{Var}(f)^* &= \frac{1}{N}\left(\langle f^2 \rangle^* - \langle f \rangle^{*2}\right) \\
&= \frac{1}{N}\mathrm{var}(f)^*,
\end{aligned}
\tag{12}
$$

where $\mathrm{var}(f)^*$ is the variance in fitness for a node. Using the probability density for fitness of a node, $p^*(f) = \frac{1}{f_{max}-f_{min}}$, we get the standard deviation of the steady-state fitness to be $\frac{f_{max}-f_{min}}{\sqrt{12N}}$.

## 5.3 Mutation rates threshold and fixation times

Here we derive the expressions for the average fixation times of a mutant, $\tau_1$ on various network topologies like the self-looped star, star, complete, cycle and the self-looped directed line.

**5.3.1 Star graphs.** To compute the fixation time for the star graph and self-looped weighted star graph, we use the method of solving recursions inspired from Ref. [54]. To proceed, we write down the recursion satisfied by $\tau_i^\bullet$, the average fixation time starting with $i$ mutants in the leaves and a mutant in the center node. We denote this state by $(\bullet, i)$. Similarly, $\tau_i^\circ$, is the average fixation time starting with the state $(\circ, i)$, i.e., $i$ mutants in the leaves and a wild-type individual in the central node.

$$
\begin{aligned}
\phi_i^\bullet \tau_i^\bullet &= T_{i,i+1}^{\bullet\bullet}\phi_{i+1}^\bullet\tau_{i+1}^\bullet + T_{i,i}^{\bullet\circ}\phi_i^\circ\tau_i^\circ + (1 - T_{i,i+1}^{\bullet\bullet} - T_{i,i}^{\bullet\circ})\phi_i^\bullet\tau_i^\bullet + \phi_i^\bullet, && 0 \le i \le n-1, \\
\phi_i^\circ \tau_i^\circ &= T_{i,i}^{\circ\bullet}\phi_i^\bullet\tau_i^\bullet + T_{i,i-1}^{\circ\circ}\phi_{i-1}^\circ\tau_{i-1}^\circ + (1 - T_{i,i}^{\circ\bullet} - T_{i,i-1}^{\circ\circ})\phi_i^\circ\tau_i^\circ + \phi_i^\circ, && 1 \le i \le n,
\end{aligned}
\tag{13}
$$

where,

  (i).  $\phi_i^\bullet$ is the fixation probability with the initial state being $(\bullet, i)$,

 (ii).  $\phi_i^\circ$ is the fixation probability with the initial state $(\circ, i)$.

(iii).  $T_{i,i\pm1}^{\bullet\bullet}$ is the transition probability from the state $(\bullet, i)$ to the state $(\bullet, i \pm 1)$,

 (iv).  $T_{i,i\pm1}^{\circ\circ}$ is the transition probability from the state $(\circ, i)$ to the state $(\circ, i \pm 1)$,

  (v).  $T_{i,i}^{\bullet\circ}$ is the transition probability from the state $(\bullet, i)$ to the state $(\circ, i)$.

 (vi).  $T_{i,i}^{\circ\bullet}$ is the transition probability from the state $(\circ, i)$ to the state $(\bullet, i)$.

The recursions in Eq 13 satisfy the boundary conditions: $\phi_0^\circ = 0$ and $\tau_n^\bullet = 0$. These recursions can be simplified further by dividing the recursion one by $T_{i,i+1}^{\bullet\bullet} + T_{i,i}^{\bullet\circ}$, and recursion two by $T_{i,i}^{\circ\bullet} + T_{i,i-1}^{\circ\circ}$,

$$
\begin{aligned}
\phi_i^\bullet \tau_i^\bullet &= \frac{T_{i,i+1}^{\bullet\bullet}}{T_{i,i+1}^{\bullet\bullet} + T_{i,i}^{\bullet\circ}}\phi_{i+1}^\bullet\tau_{i+1}^\bullet + \frac{T_{i,i}^{\bullet\circ}}{T_{i,i+1}^{\bullet\bullet} + T_{i,i}^{\bullet\circ}}\phi_i^\circ\tau_i^\circ + \frac{\phi_i^\bullet}{T_{i,i+1}^{\bullet\bullet} + T_{i,i}^{\bullet\circ}}, && 0 \le i \le n-1, \\
\phi_i^\circ \tau_i^\circ &= \frac{T_{i,i}^{\circ\bullet}}{T_{i,i}^{\circ\bullet} + T_{i,i-1}^{\circ\circ}}\phi_i^\bullet\tau_i^\bullet + \frac{T_{i,i-1}^{\circ\circ}}{T_{i,i}^{\circ\bullet} + T_{i,i-1}^{\circ\circ}}\phi_{i-1}^\circ\tau_{i-1}^\circ + \frac{\phi_i^\circ}{T_{i,i}^{\circ\bullet} + T_{i,i-1}^{\circ\circ}}, && 1 \le i \le n.
\end{aligned}
\tag{14}
$$

Introducing

$$\pi_{i,i+1}^{\bullet\bullet} = 1 - \pi_{i,i}^{\bullet\circ} = 1 - \frac{T_{i,i}^{\bullet\bullet\circ}}{T_{i,i+1}^{\bullet\bullet} + T_{i,i}^{\bullet\bullet\circ}}, \qquad 0 \le i \le n-1 \qquad (15)$$

and

$$\pi_{i,i}^{\circ\bullet} = 1 - \pi_{i,i-1}^{\circ\circ} = 1 - \frac{T_{i,i-1}^{\circ\circ}}{T_{i,i}^{\circ\bullet} + T_{i,i-1}^{\circ\circ}}, \qquad 1 \le i \le n, \qquad (16)$$

we finally have,

$$
\begin{aligned}
\phi_i^{\bullet}\tau_i^{\bullet} &= \pi_{i,i+1}^{\bullet\bullet}\phi_{i+1}^{\bullet}\tau_{i+1}^{\bullet} + \pi_{i,i}^{\bullet\circ}\phi_i^{\circ}\tau_i^{\circ} + \frac{\phi_i^{\bullet}}{T_{i,i+1}^{\bullet\bullet} + T_{i,i}^{\bullet\bullet\circ}}, \qquad 0 \le i \le n-1, \\
\phi_i^{\circ}\tau_i^{\circ} &= \pi_{i,i}^{\circ\bullet}\phi_i^{\bullet}\tau_i^{\bullet} + \pi_{i,i-1}^{\circ\circ}\phi_{i-1}^{\circ}\tau_{i-1}^{\circ} + \frac{\phi_i^{\circ}}{T_{i,i}^{\circ\bullet} + T_{i,i-1}^{\circ\circ}}, \qquad 1 \le i \le n.
\end{aligned}
\qquad (17)
$$

Here,

(i).  $\pi_{i,i+1}^{\bullet\bullet}$ is the conditional transition probability from the state $(\bullet, i)$ to the state $(\bullet, i+1)$, with the condition that the number of mutants changes.

(ii).  $\pi_{i,i}^{\circ\bullet}$ is the conditional transition probability from the state $(\circ, i)$ to the state $(\bullet, i)$, given that the number of mutants changes.

(iii).  $\pi_{i,i}^{\circ\bullet}$ is the conditional transition probability from the state $(\circ, i)$ to the state $(\bullet, i)$, with the condition that the number of mutants changes.

(iv).  $\pi_{i,i-1}^{\circ\circ}$ is the conditional transition probability from the state $(\circ, i)$ to the state $(\circ, i-1)$, given that the number of mutants changes.

Solving the recursions 17 using boundary conditions $\phi_0^{\circ} = 0$ and $\tau_n^{\bullet} = 0$ we get

$$\tau_0^{\bullet} = \tau_1^{\bullet} + 1 = \sum_{l=2}^{n} A(l,n)C(l) + 1, \qquad (18)$$

where,

$$A(l,m) = 1 + \sum_{j=l}^{m-1} \pi_{j,j}^{\bullet\circ} \prod_{k=l}^{j} \frac{\pi_{k,k-1}^{\circ\circ}}{\pi_{k,k+1}^{\bullet\bullet}} \qquad (19)$$

and

$$C(l) = \frac{\pi_{l-1,l-1}^{\bullet\circ}}{\pi_{l-1,l}^{\bullet\bullet}} \sum_{j=1}^{l-1} \left( \frac{\phi_j^{\circ}}{T_{j,j-1}^{\circ\circ} + T_{j,j}^{\circ\bullet}} \prod_{k=j+1}^{l-1} \pi_{k,k-1}^{\circ\circ} \right) + \frac{\phi_{l-1}^{\bullet}}{T_{l-1,l}^{\bullet\bullet}}. \qquad (20)$$

The expressions for $\phi_i^{\circ}$ and $\phi_i^{\bullet}$ are derived in Ref. [54],

$$
\begin{aligned}
\phi_i^{\bullet} &= \frac{A(1,i)}{A(1,n)}, \\
\phi_i^{\circ} &= \sum_{j=1}^{i} \pi_{j,j}^{\circ\bullet}\phi_j^{\bullet} \prod_{k=j+1}^{i} \pi_{k,k-1}^{\circ\circ}.
\end{aligned}
\qquad (21)
$$

Now,

$$
\begin{aligned}
\tau_1^\circ &= \tau_1^\bullet + \frac{1}{T_{1,0}^{\circ\circ} + T_{1,1}^{\circ\bullet}}, \\
&= \sum_{l=2}^{n} A(l,n)C(l) + \frac{1}{T_{1,0}^{\circ\circ} + T_{1,1}^{\circ\bullet}}.
\end{aligned}
\tag{22}
$$

The self-looped (weighted) star graph is defined by the weighted adjacency matrix

$$
\mathbf{w} = \begin{pmatrix}
1-\delta & \frac{\delta}{n} & \frac{\delta}{n} & \cdots & \frac{\delta}{n} \\
\lambda & 1-\lambda & 0 & \cdots & 0 \\
\vdots & \vdots & \ddots & \ddots & \vdots \\
\lambda & 0 & \cdots & 1-\lambda & 0 \\
\lambda & 0 & \cdots & 0 & 1-\lambda
\end{pmatrix}
\tag{23}
$$

with $0 < \lambda \leq 1$ and $0 < \delta \leq 1$. Here, $w_{ij}$ is the weight of the link directed from node $i$ to node $j$ with the center being node number 0. With this, the transition probabilities for a weighted self-looped star graph for the transitions from state $(\bullet, i)$ are

$$
T_{i,i+1}^{\bullet\bullet} = \frac{r}{r+ir+n-i} \cdot \frac{\delta}{n}(n-i), \quad \text{and} \quad T_{i,i}^{\bullet\circ} = \frac{n-i}{r+ir+n-i} \cdot \lambda.
\tag{24}
$$

The related conditional transition probabilities are

$$
\pi_{i,i+1}^{\bullet\bullet} = \frac{r\delta}{n\lambda + r\delta} \quad \text{and} \quad \pi_{i,i}^{\bullet\circ} = \frac{n\lambda}{n\lambda + r\delta}.
\tag{25}
$$

Similarly, the transition probabilities for the transitions from state $(\circ, i)$ are

$$
T_{i,i}^{\circ\bullet} = \frac{ir}{1+ir+n-i} \cdot \lambda \quad \text{and} \quad T_{i,i-1}^{\circ\circ} = \frac{1}{1+ir+n-i} \cdot \frac{\delta}{n}i.
\tag{26}
$$

The corresponding conditional transition probabilities are

$$
\pi_{i,i}^{\circ\bullet} = \frac{n\lambda r}{n\lambda r + \delta} \quad \text{and} \quad \pi_{i,i-1}^{\circ\circ} = \frac{\delta}{n\lambda r + \delta}.
\tag{27}
$$

We can use these probabilities along with Eq 22 to obtain the temperature initialised fixation probability and the average fixation time for the self-looped star graph, $\tau^{\mathcal{T}}$. In the following, we define the temperature for the center and leaf nodes. The central node temperature is

$$
\mathcal{T}_0 = \sum_{i=0}^{N} w_{i0} = 1 - \delta + n\lambda
\tag{28}
$$

and the leaf node temperature is

$$
\mathcal{T}_{j\neq0} = \sum_{i=0}^{N} w_{ij} = \frac{\delta}{n} + 1 - \lambda.
\tag{29}
$$

The temperature initialised fixation probability for the self-looped star graph is

$$
\phi^{\mathcal{T}}(\delta, \lambda) = \frac{1-\delta+n\lambda}{n+1}\phi_0^\bullet + \frac{n\left(\frac{\delta}{n}+1-\lambda\right)}{n+1}\phi_1^\circ.
\tag{30}
$$

The temperature initialised average fixation time for the self-looped star graph is

$$\tau^{\mathcal{T}}(\delta,\lambda) = \frac{1-\delta+n\lambda}{n+1}\tau_0^{\bullet} + \frac{n\left(\frac{\delta}{n}+1-\lambda\right)}{n+1}\tau_1^{\circ}. \tag{31}$$

Substituting $\lambda = \frac{1}{n}$ and $\delta = \frac{1}{n^2}$ in the above equation, we get the temperature initialised average fixation time for the self-looped weighted star graph. Setting $\lambda = \delta = 1$ yields the temperature initialised average fixation time for the standard star graph.

To compute the average extinction time, we use symmetry arguments in Eqs 18, 19, 20, 21 and, 22. With this, we replace

$$\begin{aligned}
T_{i,i+1}^{\bullet\bullet} &\text{ by } T_{n-i,n-i-1}^{\circ\circ}, \\
T_{i,i}^{\bullet\circ} &\text{ by } T_{n-i,n-i}^{\circ\bullet}, \\
T_{i,i-1}^{\circ\circ} &\text{ by } T_{n-i,n-i+1}^{\bullet\bullet}, \\
T_{i,i}^{\circ\bullet} &\text{ by } T_{n-i,n-i}^{\bullet\circ}, \\
\phi_i^{\bullet} &\text{ by } \tilde{\phi}_{n-i}^{\circ}, \text{ and,} \\
\phi_i^{\circ} &\text{ by } \tilde{\phi}_{n-i}^{\bullet}.
\end{aligned} \tag{32}$$

Doing so, we obtain

$$\tilde{\phi}_{n-i}^{\circ} = \frac{\tilde{A}(1,i)}{\tilde{A}(1,n)}, \tag{33}$$

where

$$\tilde{A}(l,m) = 1 + \sum_{j=l}^{m-1}\pi_{n-j,n-j}^{\circ\bullet}\prod_{k=l}^{j}\frac{\pi_{n-k,n-k+1}^{\bullet\bullet}}{\pi_{n-k,n-k-1}^{\circ\circ}}. \tag{34}$$

$\tilde{\phi}_i^{\circ}$ is the extinction probability of mutants starting with the state $(\circ, i)$ node, and is equal to $1 - \phi_i^{\circ}$. Similarly, the average extinction time starting with the state $(\circ, n-i)$ obeys

$$\tilde{\tau}_{n-i}^{\circ} = \sum_{l=2}^{n}\tilde{A}(l,n)\tilde{C}(l) - \frac{1}{\tilde{\phi}_{n-i}^{\circ}}\sum_{l=2}^{i}\tilde{A}(l,i)\tilde{C}(l), \tag{35}$$

where

$$\tilde{C}(l) = \frac{\tilde{\phi}_{n-l+1}^{\circ}}{T_{n-l+1,n-l}^{\circ\circ}} + \frac{\pi_{n-l+1,n-l+1}^{\circ\bullet}}{\pi_{n-l+1,n-l}^{\circ\circ}}\sum_{j=1}^{l-1}\left(\frac{\tilde{\phi}_{n-j}^{\bullet}}{T_{n-j,n-j+1}^{\bullet\bullet}+T_{n-j,n-j}^{\bullet\circ}}\prod_{k=j+1}^{l-1}\pi_{n-k,n-k+1}^{\bullet\bullet}\right), \tag{36}$$

with $\tilde{\phi}_i^{\bullet}$ being the extinction probability of mutants starting in state $(\bullet, i)$. It is given by

$$\tilde{\phi}_{n-i}^{\bullet} = \sum_{j=1}^{i}\pi_{n-j,n-j}^{\bullet\circ}\tilde{\phi}_{n-j}^{\circ}\prod_{k=j+1}^{i}\pi_{n-k,n-k+1}^{\bullet\bullet}. \tag{37}$$

The average extinction time starting in state $(\bullet, n-i)$ is

$$\tilde{\tau}_{n-i}^{\bullet} = \frac{1}{\tilde{\phi}_{n-i}^{\bullet}}\sum_{j=1}^{i}\pi_{n-j,n-j}^{\bullet\circ}\left(\tilde{\phi}_{n-j}^{\circ}\tilde{\tau}_{n-j}^{\circ} + \frac{\tilde{\phi}_{n-j}^{\bullet}}{T_{n-j,n-j}^{\bullet\circ}}\right)\prod_{k=j+1}^{i}\pi_{n-k,n-k+1}^{\bullet\bullet}. \tag{38}$$

Finally, using Eqs 38 and 35, the temperature initialised average extinction time of a mutant

on the looping star graph is

$$\tilde{\tau}^{\mathcal{T}}(\delta, \lambda) = \frac{1 - \delta + n\lambda}{n+1} \tilde{\tau}_0^\bullet + \frac{n\left(\frac{\delta}{n} + 1 - \lambda\right)}{n+1} \tilde{\tau}_1^\circ. \tag{39}$$

From Fig 6, we see that the average fixation time of a mutant is higher than the average extinction time of the mutant regardless of its relative fitness. Moreover, the fixation time peaks near neutrality. Therefore, according to the Eq 4, the $\mu_{th}$ for the stars graphs is the inverse of the average fixation times at neutrality. In the next section, we derive the scaling of $\tau_1^{\mathcal{T}}$ at neutrality with respect to the population size $N$ for the star graphs.

**5.3.2 Scaling of the average fixation time with population size for the star graphs at neutrality.** While the approach used above to compute the fixation and the extinction time on star graph has many merits like extension to the frequency dependent selection case, it is not straightforward to use this approach to derive the exact formula even at neutrality. Therefore, to compute the scaling relation for the fixation time on star graphs, we use a method inspired from Ref. [55]. To start, we recast the recursion Eq 14 into the form

$$\phi_i^\bullet \tau_i^\bullet = \pi_\rightarrow \phi_{i+1}^\bullet \tau_{i+1}^\bullet + \pi_\downarrow \phi_i^\circ \tau_i^\circ + \phi_i^\bullet t_i^\bullet, \qquad 0 \leq i \leq n-1, \tag{40}$$

$$\phi_i^\circ \tau_i^\circ = \pi_\uparrow \phi_i^\bullet \tau_i^\bullet + \pi_\leftarrow \phi_{i-1}^\circ \tau_{i-1}^\circ + \phi_i^\circ t_i^\circ, \qquad 1 \leq i \leq n, \tag{41}$$

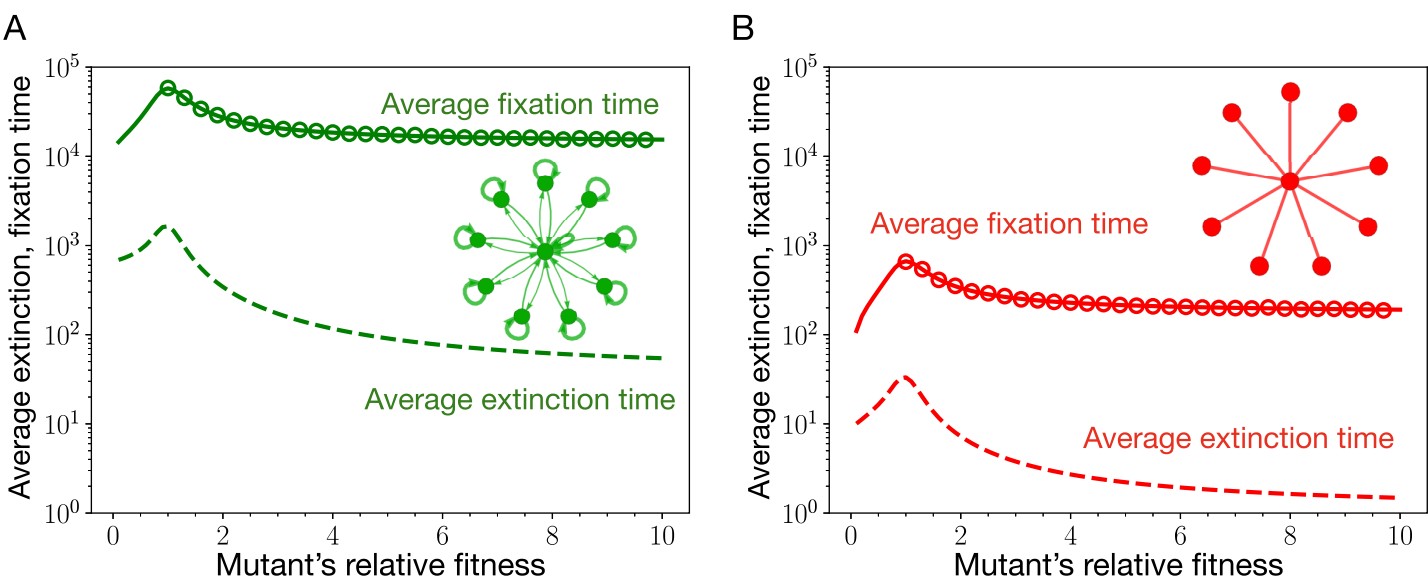

**Fig 6. Average extinction and fixation time for the self-looped star graph and the standard star graph.** Here, we plot the average extinction and fixation time of a mutant for the self-looped (weighted) star graph (panel A) and the star graph (panel B) as a function of mutant's relative fitness. Solid lines corresponds to the analytic results, Eqs 31 and 39. The circles represent Moran Bd simulations. Firstly, we observe that for both the graphs, the average fixation time of a mutant is higher than its extinction time, regardless of the mutant's relative fitness. Secondly, the average fixation time peaks near neutrality for both of the graphs. Therefore, according to Eq 4, $\mu_{th}$ for the star graphs scales as the inverse average fixation time at neutrality. Because the fixation of a mutant takes longer on the self-looped star graph, the weak mutation rate approximation is more restrictive for the self-looped star graph than the star graph. (Parameters: $N = 10$, wild-type fitness, $f = 1$, and the number of independent realisations conditioned on mutant's fixation or extinction are 2000).

Here, we have replaced,

$$
\begin{aligned}
\pi_{i,i+1}^{\bullet\bullet} &\text{ by } \pi_{\rightarrow}\,, \\
\pi_{i,i}^{\bullet\circ} &\text{ by } \pi_{\downarrow}\,, \\
\pi_{i,i}^{\circ\bullet} &\text{ by } \pi_{\uparrow}\,, \\
\pi_{i,i-1}^{\circ\circ} &\text{ by } \pi_{\leftarrow}\,,
\end{aligned}
\tag{42}
$$

because the conditional transition probabilities are independent of the number of mutants, see Eq 25. The horizontal arrows in the subscript of $\pi$ represent change in the number of mutants in the leaf nodes, right arrow for the increase, and left arrow for the decrease in the number of mutants. The vertical arrows in the subscript of $\pi$ represent change in individual type at the central node, upward arrow for the change from the wild-type to mutant type, and downward arrow for the change from the mutant type to the wild-type. We also use the shorthand notations

$$
\begin{aligned}
t_i^{\bullet} &= \frac{1}{T_{i,i+1}^{\bullet\bullet} + T_{i,i}^{\bullet\circ}}\,, \\
t_i^{\circ} &= \frac{1}{T_{i,i}^{\circ\bullet} + T_{i,i-1}^{\circ\circ}}\,.
\end{aligned}
\tag{43}
$$

Here, $t_i^{\bullet}$ is the average time spent in the state $(\bullet, i)$ (the sojourn time of state $(\bullet, i)$) and $t_i^{\circ}$ is the sojourn time of state $(\circ, i)$. Shifting the index $i$ to $i-1$ in recursion Eq 40, and solving for $\phi_i^{\bullet}\tau_i^{\bullet}$ gives,

$$
\phi_i^{\bullet}\tau_i^{\bullet} = \frac{1}{\pi_{\rightarrow}}\,\phi_{i-1}^{\bullet}\tau_{i-1}^{\bullet} - \frac{\pi_{\downarrow}}{\pi_{\rightarrow}}\,\phi_{i-1}^{\circ}\tau_{i-1}^{\circ} - \frac{1}{\pi_{\rightarrow}}\,\phi_{i-1}^{\bullet}t_{i-1}^{\bullet}.
\tag{44}
$$

Now we substitute this relation for $\phi_i^{\bullet}\tau_i^{\bullet}$ in the recursion Eq 41, and obtain,

$$
\phi_i^{\circ}\tau_i^{\circ} = \frac{\pi_{\uparrow}}{\pi_{\rightarrow}}\,\phi_{i-1}^{\bullet}\tau_{i-1}^{\bullet} + \left(\pi_{\leftarrow} - \frac{\pi_{\uparrow}\pi_{\downarrow}}{\pi_{\rightarrow}}\right)\phi_{i-1}^{\circ}\tau_{i-1}^{\circ} + \phi_i^{\circ}t_i^{\circ} - \frac{\pi_{\uparrow}}{\pi_{\rightarrow}}\,\phi_{i-1}^{\bullet}t_{i-1}^{\bullet}.
\tag{45}
$$

Recursion Eqs 44 and 45, can be written in a matrix representation as,

$$
\underbrace{\begin{bmatrix} \phi_i^{\bullet}\tau_i^{\bullet} \\ \phi_i^{\circ}\tau_i^{\circ} \end{bmatrix}}_{\mathbf{V}_i} = \underbrace{\begin{bmatrix} \frac{1}{\pi_{\rightarrow}} & -\frac{\pi_{\downarrow}}{\pi_{\rightarrow}} \\ \frac{\pi_{\uparrow}}{\pi_{\rightarrow}} & \pi_{\leftarrow} - \frac{\pi_{\uparrow}\pi_{\downarrow}}{\pi_{\rightarrow}} \end{bmatrix}}_{\mathbf{A}} \underbrace{\begin{bmatrix} \phi_{i-1}^{\bullet}\tau_{i-1}^{\bullet} \\ \phi_{i-1}^{\circ}\tau_{i-1}^{\circ} \end{bmatrix}}_{\mathbf{V}_{i-1}} + \underbrace{\begin{bmatrix} -\frac{1}{\pi_{\rightarrow}}\,\phi_{i-1}^{\bullet}t_{i-1}^{\bullet} \\ \phi_i^{\circ}t_i^{\circ} - \frac{\pi_{\uparrow}}{\pi_{\rightarrow}}\,\phi_{i-1}^{\bullet}t_{i-1}^{\bullet} \end{bmatrix}}_{\mathbf{U}_{i-1}}.
\tag{46}
$$

The matrix equation can be further simplified,

$$
\begin{aligned}
\mathbf{V}_i &= \mathbf{A}\mathbf{V}_{i-1} + \mathbf{U}_{i-1} \\
&= \mathbf{A}^i\mathbf{V}_0 + \sum_{j=0}^{i-1}\mathbf{A}^{i-j-1}\mathbf{U}_j.
\end{aligned}
\tag{47}
$$

Remember that we want to compute the scaling for $\tau^T$, and for that we need to solve the above matrix equation for $\tau_0^{\bullet}$, and $\tau_1^{\circ}$. The first thing that we need to calculate for Eq 47 is $\mathbf{A}^i$. Using

Eq 25 we substitute for the conditional probabilities in the definition of matrix $\mathbf{A}$,

$$\mathbf{A} = \begin{bmatrix} \frac{n\lambda}{\delta} + 1 & -\frac{n\lambda}{\delta} \\[2mm] \frac{n\lambda}{\delta} & 1 - \frac{n\lambda}{\delta} \end{bmatrix} = \frac{n\lambda}{\delta}\begin{bmatrix} 1 & -1 \\ 1 & -1 \end{bmatrix} + \begin{bmatrix} 1 & 0 \\ 0 & 1 \end{bmatrix} \tag{48}$$

In this way we find,

$$\mathbf{A}^i = i\frac{n\lambda}{\delta}\begin{bmatrix} 1 & -1 \\ 1 & -1 \end{bmatrix} + \begin{bmatrix} 1 & 0 \\ 0 & 1 \end{bmatrix} \tag{49}$$

To evaluate $\tau_0^{\bullet}$, we take the first row of the matrix Eq 47, and set $i = n$,

$$\phi_n^{\bullet}\tau_n^{\bullet} = \left(\frac{n^2\lambda}{\delta} + 1\right)\phi_0^{\bullet}\tau_0^{\bullet} + \sum_{j=0}^{n-1}[\mathbf{A}^{n-j-1}\mathbf{U}_j]_0, \tag{50}$$

where $\begin{bmatrix} \vdots \end{bmatrix}_0$ is the $0^{th}$ element of the column vector $\begin{bmatrix} \vdots \end{bmatrix}$. Using the boundary condition, $\tau_n^{\bullet} = 0$ in Eq 50 we find

$$\tau_0^{\bullet} = -\frac{\sum_{j=0}^{n-1}[\mathbf{A}^{n-j-1}\mathbf{U}_j]_0}{\left(\frac{n^2\lambda}{\delta} + 1\right)\phi_0^{\bullet}}. \tag{51}$$

From Refs. [54, 56] we know that at neutrality

$$\phi_i^{\bullet} = \frac{\delta + in\lambda}{\delta + n^2\lambda}, \quad \text{and} \quad \phi_i^{\circ} = \frac{in\lambda}{\delta + n^2\lambda}. \tag{52}$$

These relations for the fixation probability also follow by substituting for the transition probabilities in Eq 21. Using $\phi_0^{\bullet}$ in Eq 51, we find

$$\tau_0^{\bullet} = -\sum_{j=0}^{n-1}[\mathbf{A}^{n-j-1}\mathbf{U}_j]_0. \tag{53}$$

In order to simplify the r.h.s of the above equation, we need expressions for the waiting times in the state $i$, namely, $t_i^{\bullet}$, and $t_i^{\circ}$. We compute these expressions using Eqs 24, 26 and 43,

$$t_i^{\bullet} = \frac{n(n+1)}{(n-i)(\delta + n\lambda)} \quad \text{and} \quad t_i^{\circ} = \frac{n(n+1)}{i(\delta + n\lambda)}. \tag{54}$$

With all these expressions, we can now simplify the r.h.s of the Eq 53,

$$\sum_{j=0}^{n-1} [\mathbf{A}^{n-j-1}\mathbf{U}_j]_0 = \sum_{j=0}^{n-1} \left[ \left( (n-j-1)\frac{n\lambda}{\delta} + 1 \right) \left( -\left( \frac{n\lambda+\delta}{\delta} \right) \phi_j^\bullet t_j^\bullet \right) \right. \tag{55}$$

$$\left. -(n-j-1)\frac{n\lambda}{\delta} \left( \phi_{j+1}^\circ t_{j+1}^\circ - \frac{n\lambda}{\delta}\phi_j^\bullet t_j^\bullet \right) \right], \tag{56}$$

$$= \frac{n(n+1)}{(\delta+n\lambda)(\delta+n^2\lambda)}\sum_{j=0}^{n-1}\left[\frac{\delta+n^2\lambda}{j-n} - (n-1)\frac{n^2\lambda^2}{\delta}\right], \tag{57}$$

$$= -\frac{n(n+1)}{(\delta+n\lambda)(\delta+n^2\lambda)}\left[(\delta+n^2\lambda)H_n + (n-1)\frac{n^3\lambda^2}{\delta}\right], \tag{58}$$

where $H_n = \sum_{k=1}^{n}\frac{1}{k}$ is the harmonic number. This gives us an expression for the conditional average fixation time at neutrality on the self-looped weighted star graph starting with the state ($\bullet$, 0),

$$\tau_0^\bullet = \frac{n^4(n^2-1)}{\delta(\delta+n\lambda)(\delta+n^2\lambda)} + \frac{n(n+1)}{\delta+n\lambda}H_n. \tag{59}$$

Next, we show that $\tau_1^\circ$ is related to $\tau_0^\bullet$. To see this, let us take the second row of the matrix Eq 47, and set $i = 1$,

$$\phi_1^\circ \tau_1^\circ = \frac{\pi_\uparrow}{\pi_\rightarrow}\phi_0^\bullet\tau_0^\bullet + \left(\pi_\leftarrow - \frac{\pi_\uparrow\pi_\downarrow}{\pi_\rightarrow}\right)\underbrace{\phi_0^\circ\tau_0^\circ}_{=0} + \phi_1^\circ t_1^\circ - \frac{\pi_\uparrow}{\pi_\rightarrow}\phi_0^\bullet t_0^\bullet \tag{60}$$

Upon substituting for various quantities, we find,

$$\tau_1^\circ = \tau_0^\bullet + \frac{n^2-1}{\delta+n\lambda}. \tag{61}$$

Using the temperature initialised definition of the average fixation time for the star graph, see Eq 31, we can evaluate the expressions for the temperature initialised fixation time $\tau^\mathcal{T}$ for the self-looped star graph and the star graph without self-loops. For the self-looped star graph, setting, $\lambda = 1/n$ and $\delta = 1/n^2$, we have

$$\tau^\mathcal{T}_{\substack{\lambda=1/n, \\ \delta=1/n^2}} = \frac{(n-1)((n(n^4+n-2)+2)n^3-n+1)+(n^6+n^3)H_n}{((n-1)n+1)(n^2+1)} \tag{62}$$

$$\overset{N\gg1}{\approx} n^5 - \mathcal{O}(n^3). \tag{63}$$

For the star graph (without self-loops), setting, $\lambda = \delta = 1$, we have

$$\tau^{\mathcal{T}}_{\substack{\lambda=1,\\\delta=1}} = \frac{(n-1)(n^5+n^4+n^2+1)}{(n+1)(n^2+1)} + nH_n, \tag{64}$$

$$\stackrel{N\gg 1}{\approx} n^3 - \mathcal{O}(n^2). \tag{65}$$

**5.3.3 Gillespie algorithm.** In the Moran process, the fixation/extinction dynamics goes through many inactive steps where the configuration of the population does not change. This happens when one type is replaced by an offspring of its own type. This causes the individual-based simulation to be time-consuming, especially for large population sizes [57]. To tackle this problem, we use the Gillespie algorithm [58, 59] for the Moran fixation dynamics. We apply the Gillespie algorithm to calculate the fixation time in the star with and without self-loops for large population sizes. Simulation steps are as follows:

1. Calculate the transition probability for each possible transition which changes the configuration of the population. The possible transitions and their corresponding transition probabilities for the star graphs are discussed in the Sec. 5.3.1.

2. Calculate the total transition probability, which is the sum of all the transition probabilities. For example if the current state is $(\bullet, i)$, then the total transition probability is $T^{\bullet\bullet}_{i,i+1} + T^{\bullet\circ}_{i,i}$.

3. Generate two random numbers, one to determine the time of the next event and another to determine which event occurs. The first random number determining the time to the next event is drawn from an exponential distribution with the mean equal to the total propensity. The second random number is drawn from a uniform distribution.

4. Update the system state according to the event chosen in the previous step.

5. Repeat steps 1-4 until the system reaches fixation.

**5.3.4 Complete and cycle graph.** Compared to the star graph family, the fixation time for the complete graph (i.e. the well-mixed population) can be computed easily. From the Refs. [60–62], we know that the time to fixation for a single mutant in a population of size $N$ is given by

$$\tau_1 = \sum_{k=1}^{N-1}\sum_{l=1}^{k} \frac{\phi_l}{T_{l+}} \prod_{m=l+1}^{k} \gamma_m, \tag{66}$$

where $\gamma_m = \frac{T_{m-}}{T_{m+}}$, and $\phi_i$ is the fixation probability for mutant type to fix when started with $i$ individuals and $T_{i\pm}$ is the probability to transition from the state with $i$ mutants to the state with with $i \pm 1$ mutants. The fixation probability $\phi_i$ is given by

$$\phi_i = \frac{1 + \sum_{k=1}^{i-1}\prod_{l=1}^{k}\gamma_l}{1 + \sum_{k=1}^{N-1}\prod_{l=1}^{k}\gamma_l}. \tag{67}$$

The average fixation time on the complete when started with $i$ individuals is,

$$\tau_i = -\tau_1 \frac{\phi_1}{\phi_i} \sum_{k=i}^{N-1} \prod_{m=1}^{k} \gamma_m + \sum_{k=i}^{N-1} \sum_{l=i}^{k} \frac{\phi_l}{\phi_i} \frac{1}{T_{l+}} \prod_{m=l+1}^{k} \gamma_m. \tag{68}$$

Using symmetry arguments, similar to the ones used for the case of star graph in the previous subsection, the formula for the extinction time of $i$ mutants has been computed in Ref. [62],

$$\tilde{\tau}_i = -\tilde{\tau}_{N-1} \frac{\tilde{\phi}_{N-1}}{\tilde{\phi}_i} \sum_{k=N-i}^{N-1} \prod_{m=1}^{k} \frac{1}{\gamma_{N-m}} + \sum_{k=N-i}^{N-1} \sum_{l=1}^{k} \frac{\tilde{\phi}_{N-l}}{\tilde{\phi}_i} \frac{1}{T_{(N-l)-}} \prod_{m=l+1}^{k} \frac{1}{\gamma_{N-m}}, \tag{69}$$

where the extinction probability of $i$ mutants is $\tilde{\phi}_i = 1 - \phi_i$ and

$$\tilde{\tau}_{N-1} = \sum_{k=1}^{N-1} \sum_{l=1}^{k} \frac{\tilde{\phi}_{N-l}}{T_{(N-l)-}} \prod_{m=l+1}^{k} \frac{1}{\gamma_{N-m}}. \tag{70}$$

For the complete graph, the transition probabilities are,

$$T_{i-} = \frac{N-i}{ir+N-i} \cdot \frac{i}{N-1} \quad \text{and} \quad T_{i+} = \frac{ir}{ir+N-i} \cdot \frac{N-i}{N-1}. \tag{71}$$

These transition probabilities are plugged into Eqs 66 and 69 to obtain $\tau_1$ and $\tilde{\tau}_1$, respectively. From Fig 7A, we find that the fixation time of a mutant is higher than its extinction time regardless of its relative fitness. Moreover, the fixation time of a mutant peaks near neutrality, therefore according to Eq 4, it is the fixation time at neutrality that decides the mutation rate threshold $\mu_{th}$ for the complete graph. We now compute how the fixation time scales with $N$ for

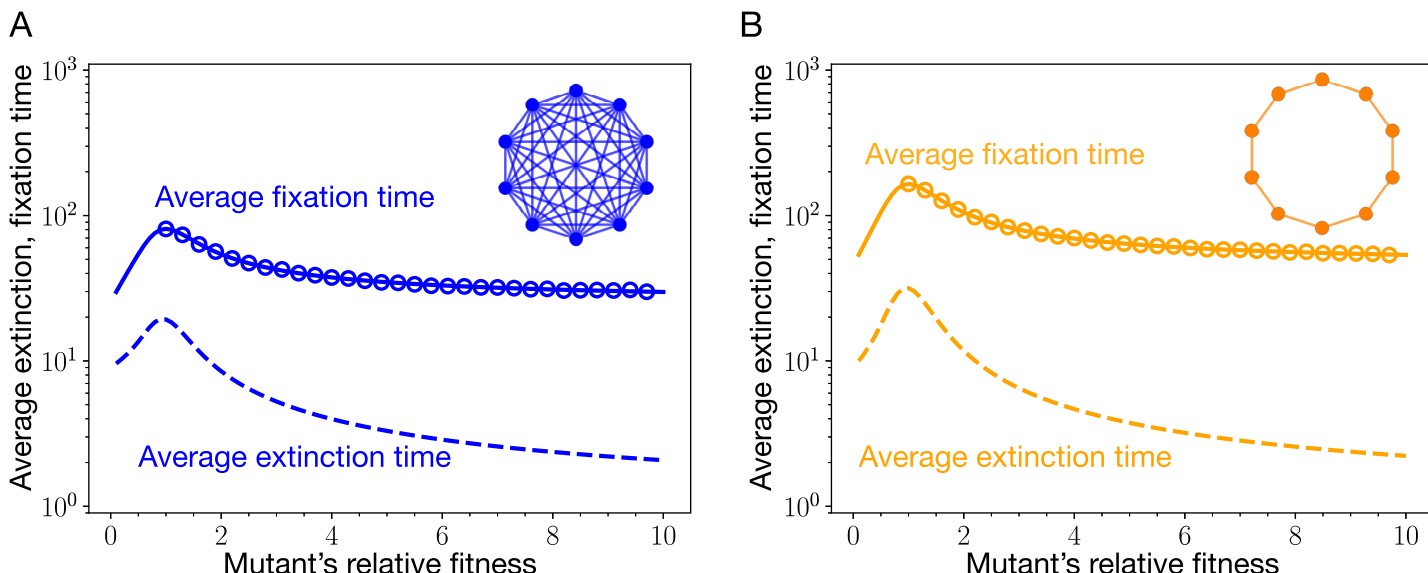

**Fig 7. Average extinction and fixation time for the isothermal graphs.** The average fixation (via solid lines) and average extinction (via dashed lines) times for the two isothermal graphs, namely, the complete graph and the cycle graph. To plot the analytical results, we have used Eqs 66 and 69. Open circles represent microscopic Moran Bd simulations. Although, the probability for a mutant to fix on any of these structures is the same due to isothermal theorem, the times it takes to reach fixation are different. Fixation on the cycle graph is slower than on the complete graph. As a result, the cycle graph is more restrictive to the weak mutation approximation. The parameters are same as in Fig 6.

the complete graph. At neutrality, $r = 1$,

$$T_{i-} = \frac{N-i}{N} \cdot \frac{i}{N-1}, \quad \text{and} \quad T_{i+} = \frac{i}{N} \cdot \frac{N-i}{N-1}. \tag{72}$$

Therefore, $\gamma_i = 1$, for all $i$. The fixation probability simplifies as to

$$\phi_i = \frac{1 + \sum_{k=1}^{i-1} \prod_{l=1}^{k} 1}{1 + \sum_{k=1}^{N-1} \prod_{l=1}^{k} 1} = \frac{i}{N}, \tag{73}$$

which is expected as every neutral mutant is equally likely to fix as any other individual of the population. Using the Eq 66, the average fixation time for a neutral mutant on the complete graph is,

$$\tau_1 = \sum_{k=1}^{N-1} \sum_{l=1}^{k} \frac{N-1}{N-l} = (N-1)^2, \tag{74}$$

which scales as $N^2$ for $N \gg 1$ [60, 63].

We now move to the cycle graph. To compute $\tau_1$ and $\tilde{\tau}_1$ for the cycle graph, the following transition probabilities are used in the Eqs 66 and 69,

$$T_{i-} = \frac{2}{ir + N - i} \cdot \frac{1}{2} \quad \text{and} \quad T_{i+} = \frac{2r}{ir + N - i} \cdot \frac{1}{2}. \tag{75}$$

Similar to the case of complete graph, from the Fig 7B, we find that the fixation time of a mutant is higher than the extinction time regardless of its relative fitness. Also, the fixation time of a mutant peaks near neutrality, the fixation time at neutrality decides the mutation rate threshold $\mu_{th}$ for the cycle graph.

Since $\gamma_m$ for the cycle graph is identical to the complete graph [10], we find the same fixation probabilities for any fitness value and initial state. In particular, at neutrality, we have $\phi_i = \frac{i}{N}$ for the cycle graph. For the cycle graph, the average fixation time for a neutral mutant is,

$$\tau_1 = \sum_{k=1}^{N-1} \sum_{l=1}^{k} l = \frac{N(N^2 - 1)}{6}, \tag{76}$$

which scales as $N^3/6$ for large $N$.

**5.3.5 Directed line with self-loops.** Here we compute the average fixation and extinction time for the self-looped directed line. Let us first study the case of fixation time. A mutant can fix on the self-looped directed line if and only if it appears at the root node. Assuming this to be the case, we have

$$T_{i-}^{\bullet} = 0 \quad \text{and} \quad T_{i+}^{\bullet} = \frac{r}{ir + N - i} \cdot \frac{1}{2}, \tag{77}$$

where $T_{i-}^{\bullet}$ is the probability to transition from the state with $i$ mutants to the state with $i + 1$ mutants given that the initial mutant appears at the root node. Similarly, $T_{i+}^{\bullet}$ is the probability to transition from the state with $i + 1$ mutants to the state with $i$ mutants given that the initial mutant appears at the root node. If the first mutant appears at the root node, the number of mutants at any time in the population can only increase. Taking this into account, we have the

average fixation time for a mutant on the directed self-looped line,

$$\tau_1 = \sum_{k=1}^{N-1} \frac{1}{T_{k+}^\bullet}, \tag{78}$$

where $1/T_{i+}^\bullet$ is average waiting time in the state with $i$ mutants, assuming that the initial mutant appeared on the root node. This expression for the $\tau_1$ can also be derived from the Eq 66. On substituting the transition probabilities $T_{i+}^\bullet$ in the r.h.s of the Eq 78, we find

$$\tau_1 = \sum_{k=1}^{N-1} \frac{2}{r}(kr + N - k) = N(N-1)\left(1 + \frac{1}{r}\right), \tag{79}$$

which scales as $N^2(1 + 1/r)$ for large $N$. Unlike other graphs, the fixation time for the self-looped directed line does not peak near neutrality, see Fig 8. Time to fixation of a mutant increases as its relative fitness decreases. Moreover, for the directed line we have $\tau_1^T = \tau_1$. In fact for any mutant initialisation scheme, the fixation time is given by the formula 79. This independency of the fixation time from the initialisation scheme holds for all the single rooted graphs.

Now, we proceed to compute the extinction time of a mutant on the self-looped directed line. However, computing the average extinction time is not as straightforward as the fixation time. Extinction takes place when an initial mutant appears on any of the non-root node. Contrary to the case of fixation where the number of mutants can only increase, here the number of mutants can increase as well as decrease. What makes things slightly complicated is that the

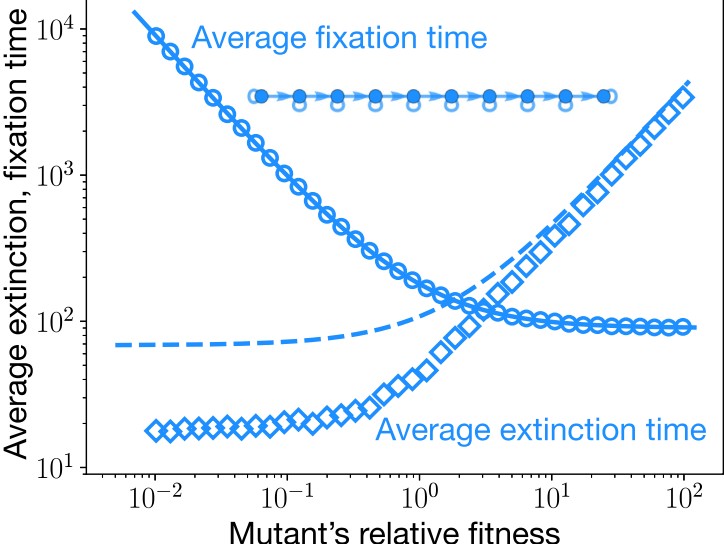

**Fig 8. Average extinction and fixation time for the self-looped directed line.** The average extinction time (dashed line) and the average fixation time (solid line) are shown for the self-looped directed line. Circles represent the total average time of the trajectories that lead to the fixation of mutants, whereas diamonds represent the average time spanned by the trajectories where mutants get extinct. We see a good agreement between analytical results and the corresponding simulations. The approximated formula for the average extinction time, Eq 82, works well in the regime of high relative fitness, as the dashed line starts coinciding with the simulations. Note that the average extinction time for a mutant can exceed the average fixation time. This is different from what we have observed in Figs 6 and 7. Also, for a given fitness domain, the average extinction and fixation time peaks away from the neutrality. Therefore, to decide the validity of the weak mutation rate approximation, fitness regions different from neutrality must be considered.

number of mutants stops to increase once the terminal node is occupied with a mutant individual. In that scenario, the transition probabilities are given as,

$$T_{i-}^{\circ} = \frac{1}{ir + N - i} \cdot \frac{1}{2} \quad \text{and} \quad T_{i+}^{\circ} = 0, \tag{80}$$

where $T_{i\pm}^{\circ}$ is the transition probability from the state with $i$ mutants to the state with $i \pm 1$ mutants, given that the initial mutant appears at a non-root node and the terminal node (node $N - 1$) of the directed line is occupied by a mutant type. We approximate the average extinction time by considering the trajectory where the number of mutants keep on increasing until the terminal node get occupied by the mutant type, and then followed by the decrease in mutants leading to extinction. An example case is shown in the Fig 9 (category third). This approximation works well in the limit of $f'/f \gg 1$. With this approximation, we have the extinction time of a mutant when appeared on a non-root node $\alpha$,

$$\tilde{\tau}_{\alpha} = N - 1 - \alpha + \sum_{k=N-\alpha}^{N-1} \frac{1}{T_{(N-k)-}^{\circ}}. \tag{81}$$

Here, $1/T_{(N-i)-}^{\circ}$ is the average waiting time in the state with $N - i$ mutants, given that the initial mutant appeared on a non-root node (node 0) and the terminal node is occupied by the mutant type. The above equation can also be derived from the Eq 69. The temperature initialised average extinction time of a mutant on the self-looped directed line is,

$$\tilde{\tau}_1^{\mathcal{T}} = \sum_{\alpha=1}^{N-1} \frac{\mathcal{T}_{\alpha}}{\sum_{\beta=1}^{N-1} \mathcal{T}_{\beta}} \tilde{\tau}_{\alpha}, \tag{82}$$

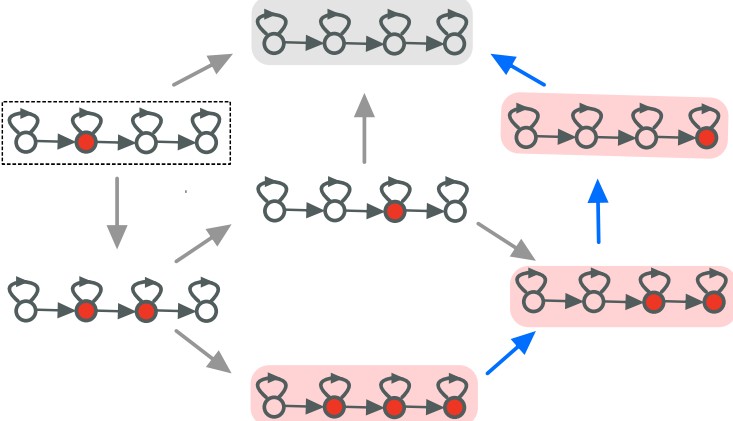

**Fig 9. Paths to extinction.** Here, the possible mutant extinction routes are shown for the self-looped directed line when the initial mutant appears on a non-root node. For purpose of illustrations, we have chosen $N = 4$. Broadly speaking, there are three categories of extinction trajectories. (i) The case where the initial mutant goes extinct without spreading in the population. This would be a one time step extinction process, shown by arrow leading from the boxed initial state to the wild-type state, highlighted in grey. (ii) The second category corresponds to the case, where the initial mutant spreads, but the mutant goes extinct before the terminal node is ever occupied by a mutant. This would contain all the paths that go from the boxed state via two mutants to the grey highlighted state without going through the states highlighted in red. (iii) The third category refers to the case, where the initial mutant spreads and reach the state highlighted in red. After the terminal node is occupied by the mutant type, the number of mutants then starts to decrease from the left (shown via the trajectory marked with blue arrows). This third category is especially relevant when the mutant's relative fitness is very high. We make use of this argument to approximate the extinction time for the self-looped directed line by computing the time covered by the blue arrowed trajectory.

where $\mathcal{T}_\alpha$ is the temperature of node $\alpha$. $\mathcal{T}_\alpha / \sum_{\beta=1}^{N-1} \mathcal{T}_\beta$ is the probability that the initial mutant appears at the node $\alpha$, given that the mutant ultimately goes extinct, i.e., given that it appears at a non-root node. From the Fig 8, we see that the average extinction time of a mutant increases with relative fitness. However, for a given fitness domain, we find that the contribution to the mutation threshold $\mu_{th}$ comes from the average fixation time computed for the lowest possible mutant's relative fitness.

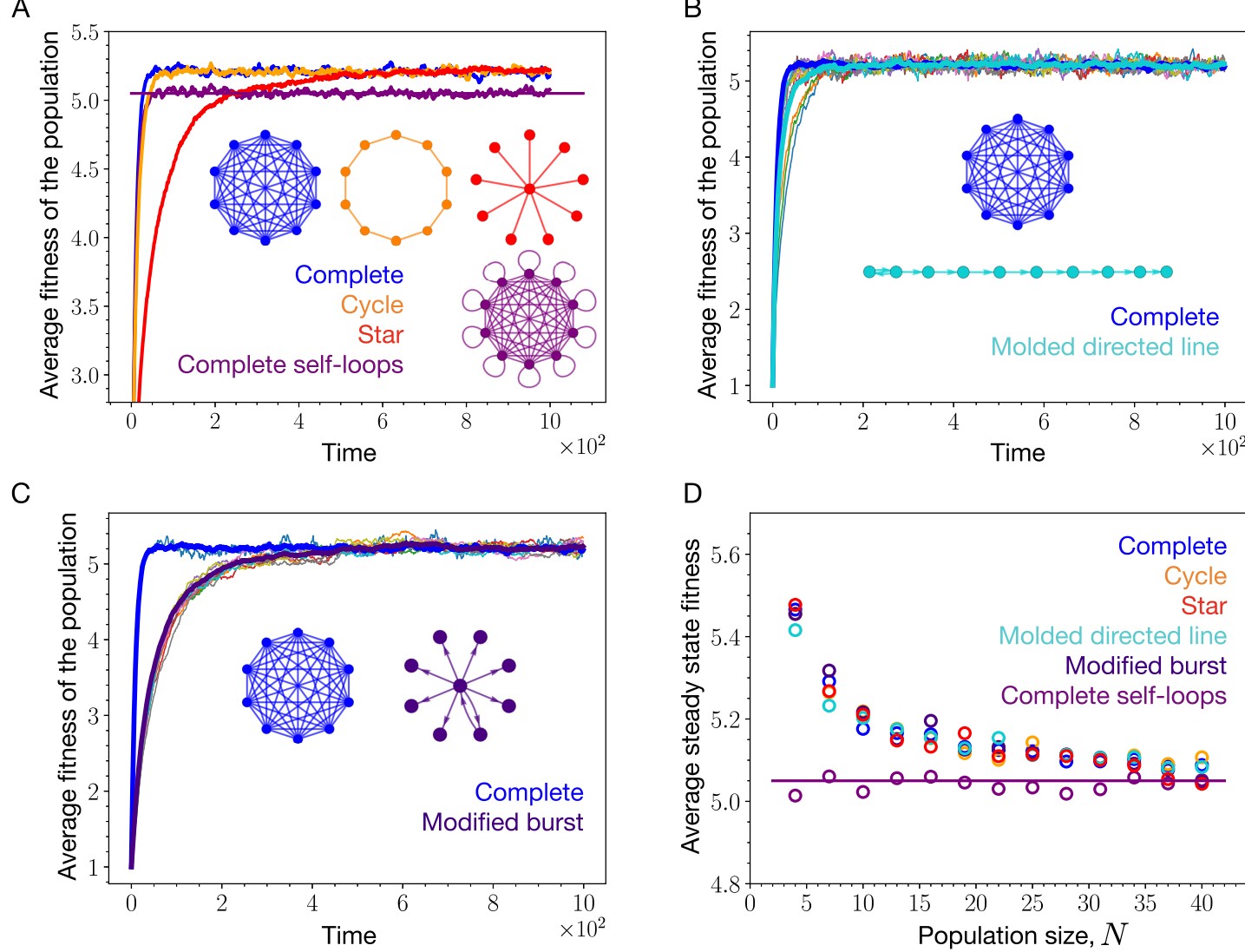

**Fig 10. Universal steady-state fitness among non self-looped graphs.** (A) In the steady-state, the complete graph, the cycle graph, and the star graph, attain the same average fitness in the mutation-selection balance. The steady-state average fitness obtained by these graphs is higher than that of the self-looped complete graph, indicating that the dynamics on these graphs—unlike the self-looped complete graph—is not entirely uncorrelated in the fitness space and time. One common thing in these three graphs is that every node has at least one incoming link. (B) The molded directed line is constructed by adding a link directed from node 1 to node 0, the root node, so that every node has non-zero incoming links. In the steady-state, not only the molded directed line attains the same average fitness as the complete graph (non self-looped), but every single node becomes indistinguishable. (C) The same observation is made for the modified burst graph. (D) These observations remain valid for different population sizes. The difference between the steady-state average fitness of the non self-looped and the self-looped complete graph decreases with increasing $N$, indicating that the evolutionary dynamics becomes more random with increasing $N$ (Parameters: same as Fig 3).

### 5.4 Non self-looped graphs and universality at equilibrium

Here, we study the evolutionary dynamics, considering both directed as well as undirected graphs, but without self-loops.

In the absence of self-loops, the average steady-state fitness for all the graphs, whether directed or not, is the same, see Fig 10A. We hypothesise that all the graphs where every node has a finite temperature attain the same steady-steady fitness in the mutation-selection balance. All the nodes for these graphs become indistinguishable in their fitness in the steady-state. The steady-state fitness attained by these graphs in the mutation-selection balance is higher than the self-looped complete graph. For these graphs, the fitnesses of nodes in the steady-state are not completely uncorrelated in time as opposed to nodes of the self-looped complete graph where the fitness states of nodes are entirely uncorrelated in time. To test our hypothesis, we analysed a few variants of the directed line and burst graph in Fig 10B and 10C, where each node has a finite temperature. To achieve this, we add a link from node 1 to the root node of the directed line yielding a molded directed line. Similarly, the modified burst is constructed by adding a link from the leaf node to the center in a burst graph. In both the cases, we find that the steady-state fitness attained by these two variant graphs in the mutation-selection balance is the same as that of the complete graph and hence, the other non-self looped graphs considered in Fig 10A. We further check the validity of our hypothesis by varying the population size. In Fig 10D, we see that all the non self-looped graph attain the same steady-state fitness for all population sizes.

Another interesting observation is that the steady-state fitness balance of the non self-looped graphs decreases with increasing $N$. This is interesting, because with increasing $N$ one typically expects that the associated increase in the selection strength leads to an increase in the fitness. However, the opposite is seen here. A possible explanation for this is that in the Moran Birth-death update scheme, high fitness nodes are more likely to be selected for reproduction, but since there are no self-loops, highly fit individuals can persist in the population for longer times than the low fitness nodes which eventually leads to an the increase in population fitness. However, with the increase in the population size, the high fitness nodes tend to get replaced relatively more often as they get less selected for birth. On increasing $N$, the steady-state average fitness of the non self-looped graphs get closer to that of the average steady-state fitness of the self-looped complete graph. In Sec. 3.4.1, we saw that at long times the fitness of nodes for the self-looped complete graph becomes uncorrelated in time and hence the dynamics becomes completely random. With the increase in $N$, the strength of randomness increases over selection of high fitness valued individuals.

## Acknowledgments

We are thankful to Julien Dutheil, Christian Hilbe and Javier Lopez Garrido for helpful discussions.

## Author Contributions

**Conceptualization:** Nikhil Sharma, Arne Traulsen.

**Formal analysis:** Nikhil Sharma, Sedigheh Yagoobi.

**Investigation:** Nikhil Sharma.

**Methodology:** Nikhil Sharma.

**Software:** Nikhil Sharma.

**Validation:** Nikhil Sharma, Sedigheh Yagoobi, Arne Traulsen.

**Visualization:** Nikhil Sharma, Arne Traulsen.

**Writing – original draft:** Nikhil Sharma.

**Writing – review & editing:** Nikhil Sharma, Sedigheh Yagoobi, Arne Traulsen.

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
