## [Decision Letter · Decision Letter 0]

21 Jun 2023

Dear Dr. Traulsen,

Thank you very much for submitting your manuscript "Self-loops in Evolutionary Graph Theory: Friends or Foes?" for consideration at PLOS Computational Biology. As with all papers reviewed by the journal, your manuscript was reviewed by members of the editorial board and by several independent reviewers. The reviewers appreciated the attention to an important topic. Based on the reviews, we are likely to accept this manuscript for publication, providing that you modify the manuscript according to the review recommendations.

Sincerely,

Yamir Moreno

Academic Editor

PLOS Computational Biology

Natalia Komarova

Section Editor

PLOS Computational Biology

Reviewer's Responses to Questions

**Comments to the Authors:**

Reviewer #1: The paper applies the high mutation regime to evolutionary graph theory and, in particular, looks at graphs with self-loops where this regime is pertinent. Graphs that are commonly studied have been considered in this paper and therefore providing a foundational basis for the high mutation regime.

I recommend this paper be accepted for publication.

Reviewer #2: The authors study some idealized structures through simulation using the Gillespie algorithm for efficiency. The main focus is on mutation processes that are uniform within some bounded space, immediately forgetting the fitness of the parent type. It is later explored briefly with a localized/Gaussian mutation process, where the qualitative findings are shown to be similar. The limit of frequent mutation (\\mu\\to 1) is emphasized here as providing contrast to the standard low-mutation assumption (\\mu\\to 0). One takeaway is that in contrast to the low-mutation case, self-loops decrease the mean fitness (relative to the complete graph) when mutations are frequent.

I find this result interesting a suitable for PLOS Computational Biology because it paints a picture that is different from what many other studies suggest, which is important in building our understanding of the role of population structure—especially because mutation rates in real populations need not be as small as required by studies of fixation probabilities alone. I also agree with the authors that amplifiers and suppressors are over-emphasized in the literature, and I think these results, although for a limited range of structures, give some important insights into dynamics one can see under realistic assumptions on parameters such as mutation rates.

I have some (mostly minor) comments that I would ask the authors to think about before publication in PLOS Computational Biology.

Comments:

-general comment: the notation \\rho for the mutation kernel might be confusing to many readers because \\rho is standard notation for fixation probabilities. this confusion comes up e.g. in eq. 1 where both fixation probabilities and mutations are present. could this be changed?

-abstract: “…that speeds up…” -> “…that speed up...”

-abstract: already I was a little confused about how amplifiers are defined here, since usually they are based on fixation probabilities rather than times (and “speed” seems to indicate time—at least at this point in the paper).

-section 1, line 40: “A suppressors of selection…” -> “Suppressors of selection…”

-section 2, line 63: please state that the mutation occurs in the offspring so that this is not confused with mutant initialization. (at present it reads almost as if each node is equally likely to have a mutant arise in it.)

-section 2, line 65: make clear that fitness is constant (no game).

-section 2, lines 67-70: it would be helpful at this point to explicitly say which mutation kernels are of interest. at this stage in the manuscript, it can be presumed only that the authors are interested in mutations appearing in some (continuous) interval.

-section 2, lines 75-79: point taken, but please emphasize that the topic of this paper is not a metapopulation model.

-section 3, eq.1: here it would be helpful to describe this in a bit more detail. the underlying intuition is an embedded Markov chain on the all-f states over all f in the space. a mutant appears with some probability, then fixes with another probability, which gives these transitions. even informally, the derivation of this equation should be recalled here.

-section 4, line 111 (and elsewhere): the “weak mutation rate regime” is mentioned several times, and it is only alluded to that this corresponds to the case in which fixation of a mutant happens more quickly than the appearance of another mutant. (a bit later, this is mentioned more, but earlier would be useful.) several studies have approached this question formally, including at least one by the last author. it would be useful for readers to have a bit of a discussion to this here, together with relevant pointers to the literature.

-section 5, line 175: this interpretation of loops “decreasing” fitness is interesting, and I’m wondering whether more can be said about what this means. For example, I assume “fitness” here doesn’t directly refer to the parameter f because this only quantifies propensity to reproduce (regardless of where the offspring go). Self-loops increase the death probability of the individual at a location with fitness f, but they can also result in fitter individuals being placed there (as well as less fit individuals). This is only briefly described later starting on line 251 for the example there.

-section 5.1.1 (line number missing): to rephrase, the choice of reference graph is to avoid situations in which those with higher fitness are less likely to be replaced by a mutant, right?

-section 5.1.1, eq. 5: since this is an ODE and not a PDE, please replace partial notion by ‘d’.

-section 5, line 230: where does the factor of \\sqrt{12} come from?

-section 5, fig. 4 caption: missing close parenthesis after “panel B”. also \\lambda is not introduced/defined.

-one more general comment that I wanted to verify: in the low-mutation case, when comparing the effects of self-loops, is the reference graph typically a well-mixed population with or without self-loops?

Reviewer #3: [see also the marked-up pdf]

Evolutionary graph theory studies evolution of spatially structured populations.

Past research has pointed to beneficial effects of self-loops, but those findings rely on and assumption that new mutations occur rarely. Here the authors look into the effect of self-loops in the regime when new mutations occur at moderate or high rates. By considering 4 specific graph topologies (complete graph, star, cycle, directed path) they show that outside of the limit of rare mutations, the effect of self-loops is detrimental rather than beneficial.

I like the content of this work. The question is natural and the results are sound. There is a mix of simulation results and analytical results (e.g. the authors prove that the fixation time on the star graph under neutral drift is of the order of N^3). Moreover, the authors give intuition about why certain simulations turned out one way or another.

I have three comments. I believe all of them can be addressed, and in that case I support ultimate publication.

1. The wording is sometimes confusing / ambiguous and the grammar sometimes appears to be wrong. Please see the marked-up pdf.

2. Occasionally (e.g. lines 90-91, or 107-108), there are sentences that claim something, but it is not clear to me as a reader whether:

- the claim follows from earlier work, or

- the claim is obvious, or

- the claim is what the authors will prove in this work, or

- the claim is what the simulations indicate (but we don't know whether it is actually true), etc

Please see the marked-up pdf and clarify.

3. Several times throughout the text, the authors refer to Ref.[20] by saying something like "for amplification under temperature initialization, self-loops are necessary". Formally speaking, this is not true: Ref [20] shows that self-loops are necessary to generate *substantial* amplification, but certain temperature-amplifiers without self-loops actually do exist, see https://doi.org/10.1371/journal.pcbi.1008695 Please clarify this.

**Have the authors made all data and (if applicable) computational code underlying the findings in their manuscript fully available?**

Reviewer #1: Yes

Reviewer #2: None

Reviewer #3: Yes

PLOS authors have the option to publish the peer review history of their article (what does this mean?). If published, this will include your full peer review and any attached files.

Reviewer #1: No

Reviewer #2: No

Reviewer #3: **Yes: **Josef Tkadlec

Figure Files:

Data Requirements:

Reproducibility:

References:

---

## [Decision Letter · Decision Letter 1]

25 Jul 2023

Dear Dr. Traulsen,

We are pleased to inform you that your manuscript 'Self-loops in Evolutionary Graph Theory: Friends or Foes?' has been provisionally accepted for publication in PLOS Computational Biology.

Best regards,

Yamir Moreno

Academic Editor

PLOS Computational Biology

Natalia Komarova

Section Editor

PLOS Computational Biology

Reviewer's Responses to Questions

**Comments to the Authors:**

Reviewer #2: Thanks for the revisions. I am happy with the current paper and recommend publication.

**Have the authors made all data and (if applicable) computational code underlying the findings in their manuscript fully available?**

Reviewer #2: None

PLOS authors have the option to publish the peer review history of their article (what does this mean?). If published, this will include your full peer review and any attached files.

Reviewer #2: No

---

## [Editor Report · Acceptance letter]

23 Aug 2023

PCOMPBIOL-D-23-00464R1 

Self-loops in Evolutionary Graph Theory: Friends or Foes?

Dear Dr Traulsen,

I am pleased to inform you that your manuscript has been formally accepted for publication in PLOS Computational Biology. Your manuscript is now with our production department and you will be notified of the publication date in due course.

With kind regards,

Zsofi Zombor
